# Contrastive Training of Complex-Valued Autoencoders for Object Discovery

**Aleksandar Stanić**[1][*] **Anand Gopalakrishnan**[1][*] **Kazuki Irie**[2][†] **Jürgen Schmidhuber**[1,3]

[1]The Swiss AI Lab, IDSIA, USI & SUPSI, Lugano, Switzerland
[2]Center for Brain Science, Harvard University, Cambridge, USA
[3]AI Initiative, KAUST, Thuwal, Saudi Arabia
{aleksandar, anand, juergen}@idsia.ch kirie@fas.harvard.edu

## Abstract

Current state-of-the-art object-centric models use slots and attention-based routing for binding. However, this class of models has several conceptual limitations: the number of slots is hardwired; all slots have equal capacity; training has high computational cost; there are no object-level relational factors within slots. Synchrony-based models in principle can address these limitations by using complex-valued activations which store binding information in their phase components. However, working examples of such synchrony-based models have been developed only very recently, and are still limited to toy grayscale datasets and simultaneous storage of less than three objects in practice. Here we introduce architectural modifications and a novel contrastive learning method that greatly improve the state-of-the-art synchrony-based model. For the first time, we obtain a class of synchrony-based models capable of discovering objects in an unsupervised manner in multi-object color datasets and simultaneously representing more than three objects.[1]

## 1 Introduction

The visual binding problem [1, 2] is of great importance to human perception [3] and cognition [4–6]. It is fundamental to our visual capabilities to integrate several features together such as color, shape, texture, etc., into a unified whole [7]. In recent years, there has been a growing interest in deep learning models [8–19] capable of grouping visual inputs into a set of 'meaningful' entities in a fully unsupervised fashion (often called object-centric learning). Such compositional object-centric representations facilitate relational reasoning and generalization, thereby leading to better performance on downstream tasks such as visual question-answering [20, 21], video game playing [22–26], and robotics [27–29], compared to other monolithic representations of the visual input.

The current mainstream approach to implement such object binding mechanisms in artificial neural networks is to maintain a set of separate activation vectors (so-called *slots*) [15]. Various segregation mechanisms [30] are then used to route requisite information from the inputs to infer each slot in a iterative fashion. Such slot-based approaches have several conceptual limitations. First, the binding information (i.e., addresses) about object instances are maintained only by the constant number of slots—a hard-wired component which cannot be adapted through learning. This restricts the ability of slot-based models to flexibly represent varying number of objects with variable precision without tuning the slot size, number of slots, number of iterations, etc. Second, the inductive bias used by the grouping module strongly enforces independence among all pairs of slots. This restricts individual

---

[*]Equal contribution.

[†]Work done at IDSIA.

[1]Official code repository: `https://github.com/agopal42/ctcae`

37th Conference on Neural Information Processing Systems (NeurIPS 2023).

slots to store relational features at the object-level, and requires additional processing of slots using a relational module, e.g., Graph Neural Networks [31, 32] or Transformer models [33–35]. Third, binding based on iterative attention is in general computationally very demanding to train [36]. Additionally, the spatial broadcast decoder [37] (a necessary component in these models) requires multiple forward/backward passes to render the slot-wise reconstruction and alpha masks, resulting in a large memory overhead as well.

Recently, Löwe et al. [36] revived another class of neural object binding models [38–40] (*synchrony-based* models) which are based on complex-valued neural networks. Synchrony-based models are conceptually very promising. In principle they address most of the conceptual challenges faced by slot-based models. The binding mechanism is implemented via constructive or destructive phase inter-ference caused by addition of complex-valued activations. They store and process information about object instances in the phases of complex activations which are more amenable to adaptation through gradient-based learning. Further, they can in principle store a variable number of objects with variable precision by partitioning the phase components of complex activations at varying levels of granularity. Additionally, synchrony-based models can represent relational information directly in their distributed representation, i.e., distance in phase space yields an implicit relational metric between object in-stances (e.g., inferring part-whole hierarchy from distance in "tag" space [39]). Lastly, the training of synchrony-based models is computationally more efficient by two orders of magnitude [36].

However, the true potential of synchrony-based models for object binding is yet to be explored; the current state-of-the-art synchrony-based model, the Complex-valued AutoEncoder (CAE) [36], still has several limitations. First, it is yet to be benchmarked on any multi-object datasets [41] with color images (even simplisitic ones like `Tetrominoes`) due to limitations in the evaluation method to extract discrete object identities from continuous phase maps [36]. Second, we empirically observe that it shows low *separability* (Table 2) in the phase space, thereby leading to very poor (near chance-level) grouping performance on `dSprites` and `CLEVR`. Lastly, CAE can simultaenously represent at most 3 objects [36], making it infeasible for harder benchmark datasets [41, 42].

Our goal is to improve the state-of-art synchrony models by addressing these limitations of CAE [36]. First, we propose a few simple architectural changes to the CAE: i) remove the `1x1` convolution kernel as well as the sigmoid activation in the output layer of decoder, and ii) use convolution and upsample layers instead of transposed convolution in the decoder. These changes enable our improved CAE, which we call CAE++, to achieve good grouping performance on the `Tetrominoes` dataset—a task on which the original CAE completely fails. Further, we introduce a novel contrastive learning method to increase *separability* in phase values of pixels (regions) belonging to two different objects. The result-ing model, which we call Contrastively Trained Complex-valued AutoEncoders (CtCAE), is the first kind of synchrony-based object binding models to achieve good grouping performance on multi-object color datasets with more than three objects (Figure 2). Our contrastive learning method yields signifi-cant gains in grouping performance over CAE++, consistently across three multi-object color datasets (`Tetrominoes`, `dSprites` and `CLEVR`). Finally, we qualitatively and quantitatively evaluate the *sep-arability* in phase space and generalization of CtCAE w.r.t. number of objects seen at train/test time.

## 2 Background

We briefly overview the CAE architecture [36] which forms the basis of our proposed models. CAE performs binding through complex-valued activations which transmit two types of messages: *magnitudes* of complex activations to represent the strength of a feature and *phases* to represent which features must be processed together. The constructive or destructive interference through addition of complex activations in every layer pressurizes the network to use similar phase values for all patches belonging to the same object while separating those associated with different objects. Patches of the same object contain a high amount of pointwise mutual information so their destructive interference would degrade its reconstruction.

The CAE is an autoencoder with *real-valued* weights that manipulate *complex-valued* activations. Let $h$ and $w$ denote positive integers. The input is a positive real-valued image $\mathbf{x}' \in \mathbb{R}^{h \times w \times 3}$ (height $h$ and width $w$, with 3 channels for color images). An artificial initial phase of zero is added to each pixel of $\mathbf{x}'$ (i.e., $\boldsymbol{\phi} = \mathbf{0} \in \mathbb{R}^{h \times w \times 3}$) to obtain a complex-valued input $\mathbf{x} \in \mathbb{C}^{h \times w \times 3}$:

$$\mathbf{x} = \mathbf{x}' \odot e^{i\boldsymbol{\phi}}, \quad \text{where } \odot \text{ denotes a Hadamard product} \tag{1}$$

Let $d_{\text{in}}$, $d_{\text{out}}$ and $p$ denote positive integers. Every layer in the CAE transforms complex-valued input $\mathbf{x} \in \mathbb{C}^{d_{\text{in}}}$ to complex-valued output $\mathbf{z} \in \mathbb{C}^{d_{\text{out}}}$ (where we simply denote input/output sizes as $d_{\text{in}}$ and $d_{\text{out}}$ which typically have multiple dimensions, e.g., $h \times w \times 3$ for the input layer), using a function $f_{\mathbf{w}} : \mathbb{R}^{d_{\text{in}}} \to \mathbb{R}^{d_{\text{out}}}$ with real-valued trainable parameters $\mathbf{w} \in \mathbb{R}^p$. $f_{\mathbf{w}}$ is typically a convolutional or linear layer. First, $f_{\mathbf{w}}$ is applied separately to the real and imaginary components of the input:

$$\boldsymbol{\psi} = f_{\mathbf{w}}\left(\text{Re}(\mathbf{x})\right) + f_{\mathbf{w}}\left(\text{Im}(\mathbf{x})\right) i \quad \in \mathbb{C}^{d_{\text{out}}} \tag{2}$$

Note that both $\text{Re}(\mathbf{x}), \text{Im}(\mathbf{x}) \in \mathbb{R}^{d_{\text{in}}}$. Second, separate trainable bias vectors $\mathbf{b_m}, \mathbf{b_\phi} \in \mathbb{R}^{d_{\text{out}}}$ are applied to the magnitude and phase components of $\boldsymbol{\psi} \in \mathbb{C}^{d_{\text{out}}}$:

$$\mathbf{m}_{\boldsymbol{\psi}} = |\boldsymbol{\psi}| + \mathbf{b_m} \quad \in \mathbb{R}^{d_{\text{out}}} \quad ; \quad \boldsymbol{\phi}_{\boldsymbol{\psi}} = \arg(\boldsymbol{\psi}) + \mathbf{b_\phi} \quad \in \mathbb{R}^{d_{\text{out}}} \tag{3}$$

Third, the CAE uses an additional gating function proposed by Reichert and Serre [40] to further transform this "intermediate" magnitude $\mathbf{m}_{\boldsymbol{\psi}} \in \mathbb{R}^{d_{\text{out}}}$. This gating function dampens the response of an output unit as a function of the phase difference between two inputs. It is designed such that the corresponding response curve approximates experimental recordings of the analogous curve from a Hodgkin-Huxley model of a biological neuron [40]. Concretely, an intermediate activation vector $\boldsymbol{\chi} \in \mathbb{R}^{d_{\text{out}}}$ (called *classic term* [40]) is computed by applying $f_{\mathbf{w}}$ to the magnitude of the input $\mathbf{x} \in \mathbb{C}^{d_{\text{in}}}$, and a convex combination of this *classic term* and the magnitude $\mathbf{m}_{\boldsymbol{\psi}}$ (called *synchrony term* [40]) from Eq. 3 is computed to yield "gated magnitudes" $\mathbf{m_z} \in \mathbb{R}^{d_{\text{out}}}$ as follows:

$$\boldsymbol{\chi} = f_{\mathbf{w}}\left(|\mathbf{x}|\right) + \mathbf{b_m} \quad \in \mathbb{R}^{d_{\text{out}}} \quad ; \quad \mathbf{m_z} = \frac{1}{2}\mathbf{m}_{\boldsymbol{\psi}} + \frac{1}{2}\boldsymbol{\chi} \quad \in \mathbb{R}^{d_{\text{out}}} \tag{4}$$

Finally, the output of the layer $\mathbf{z} \in \mathbb{C}^{d_{\text{out}}}$ is obtained by applying non-linearities to this magnitude $\mathbf{m_z}$ (Eq. 4) while leaving the phase values $\boldsymbol{\phi}_{\boldsymbol{\psi}}$ (Eq. 3) untouched:

$$\mathbf{z} = \text{ReLU}(\text{BatchNorm}(\mathbf{m_z})) \odot e^{i\boldsymbol{\phi}_{\boldsymbol{\psi}}} \quad \in \mathbb{C}^{d_{\text{out}}} \tag{5}$$

The ReLU activation ensures that the magnitude of $\mathbf{z}$ is positive, and any phase flips are prevented by its application solely to the magnitude component $\mathbf{m_z}$. For more details of the CAE [36] and gating function [40] we refer the readers to the respective papers. The final object grouping in CAE is obtained through K-means clustering based on the phases at the output of the decoder; each pixel is assigned to a cluster corresponding to an object [36].

## 3 Method

We describe the architectural and contrastive training details used by our proposed CAE++ and CtCAE models respectively below.

**CAE++.** We first propose some simple but crucial architectural modifications that enable the vanilla CAE [36] to achieve good grouping performance on multi-object datasets such as `Tetrominoes` with color images. These architectural modifications include — i) Remove the `1x1` convolution kernel and associated sigmoid activation in the output layer ("$f_{\text{out}}$" in Löwe et al. [36]) of the decoder, ii) Use convolution and upsample layers in place of transposed convolution layers in the decoder (cf. "$f_{\text{dec}}$" architecture in Table 3 from Löwe et al. [36]). We term this improved CAE variant that adopts these architectural modifications as CAE++. As we will show below in Table 1, these modifications allow our CAE++ to consistently outperform the CAE across all 3 multi-object datasets with color images.

**Contrastive Training of CAEs.** Despite the improved grouping of CAE++ compared to CAE, we still empirically observe that CAE++ shows poor *separability*[2] (we also illustrate this in Section 4). This motivates us to introduce an auxiliary training objective that explicitly encourages higher *separability* in phase space. For that, we propose a contrastive learning method [43, 44] that modulates the distance between pairs of distributed representations based on some notion of (dis)similarity between them (which we define below). This design reflects the desired behavior to drive the phase separation process between two different objects thereby facilitating better grouping performance.

Before describing our contrastive learning method mathematically below, here we explain its essential ingredients (illustrated in Figure 1). For setting up the contrastive objective, we first (randomly)

---

[2]We define separability as the minimum angular distance of phase values between a pair of prototypical points (centroids) that belong to different objects

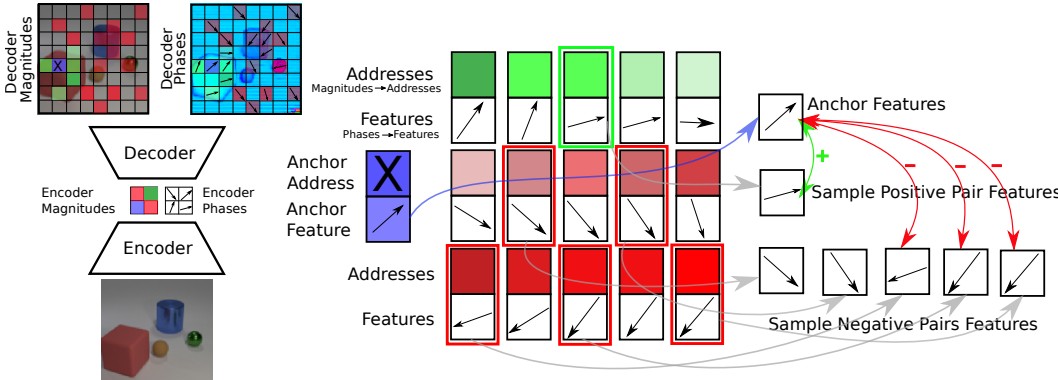

Figure 1: Sampling process of positive (green) and negative (red) pairs for one anchor (purple) in the CtCAE model. The sampling process here is visualized only for the decoder output. Note that we contrast the encoder output in an identical manner. Anchor address (the purple box marked with an X) corresponds to the patch of magnitude values and the feature corresponds to the phase values. See **Contrastive Training of CAEs** in Section 3 for more details.

---

**Algorithm 1** Mining positive and negative pairs for a single anchor for the contrastive objective. Scalar parameters are $k_{\text{top}}$ — the number of candidates from which to sample one positive pair and $m_{\text{bottom}}$ — the number of candidates from which to sample $(M-1)$ negative pairs.

---

**Inputs:** `anchor_address` $\in \mathbb{R}^{d_f}$ , `addresses` $\in \mathbb{R}^{h' \times w' \times d_f}$ , `features` $\in \mathbb{R}^{h' \times w' \times d_f}$ .
**Params:** Scalars $k_{\text{top}}, m_{\text{bottom}}, M$.
```
 1: addresses ← Flatten(addresses, dims=(0,1))
 2: features ← Flatten(features, dims=(0,1))
 3: distances ← CosineDistance(addresses, anchor_address)
 4: top_k_features ← features[argsort(distances, dim=0)[:k_top]]
 5: bottom_m_features ← features[argsort(distances, dim=0)[−m_bottom:]]
 6: pos_pair_idx ∼ Uniform [0, k_top]                         ▷ sample 1 positive pair
 7: neg_pair_idxs ∼ Uniform [0, m_bottom]×(M − 1) ▷ M − 1 samples without replacement
 8: positive_pair ← top_k_features[pos_pair_idx]
 9: negative_pairs ← bottom_m_features[neg_pair_idxs]
10: return positive_pair, negative_pairs
```

---

sample "anchors" from a set of "datapoints". The main idea of contrastive learning is to "contrast" these anchors to their respective "positive" and "negative" examples in a certain representation space. This requires us to define two representation spaces: one associated with the similarity measure to define positive/negative examples given an anchor, and another one on which we effectively apply the contrastive objective, itself defined as a certain distance function to be minimized. We use the term *addresses* to refer to the representations used to measure similarity, and consequently extract positive and negative pairs w.r.t. to the anchor. We use the term *features* to refer to the representations that are contrasted. As outlined earlier, the goal of the contrastive objective is to facilitate *separability* of phase values. It is then a natural choice to use the phase components of complex-valued outputs as *features* and the magnitude components as *addresses* in the contrastive loss. This results in angular distance between phases (*features*) being modulated by the contrastive objective based on how (dis)similar their corresponding magnitude components (*addresses*) are. Since the magnitude components of complex-valued activations are used to reconstruct the image (Equation (7)), they capture requisite visual properties of objects. In short, the contrastive objective increases or decreases the angular distance of phase components of points (pixels/image regions) in relation to how (dis)similar their visual properties are.

Our contrastive learning method works as follows. Let $h'$, $w'$, $d_f$, $N_A$, and $M$ denote positive integers. Here we generically denote the dimension of the output of any CAE layer as $h' \times w' \times d_f$. This results in a set of $h' \times w'$ "datapoints" of dimension $d_f$ for our contrastive learning. From this set of datapoints, we randomly sample $N_A$ anchors. We denote this set of anchors as a matrix $\mathbf{A} \in \mathbb{R}^{N_A \times d_f}$; each anchor is thus denoted as $\mathbf{A}_k \in \mathbb{R}^{d_f}$ for all $k \in \{1, ..., N_A\}$. Now by using

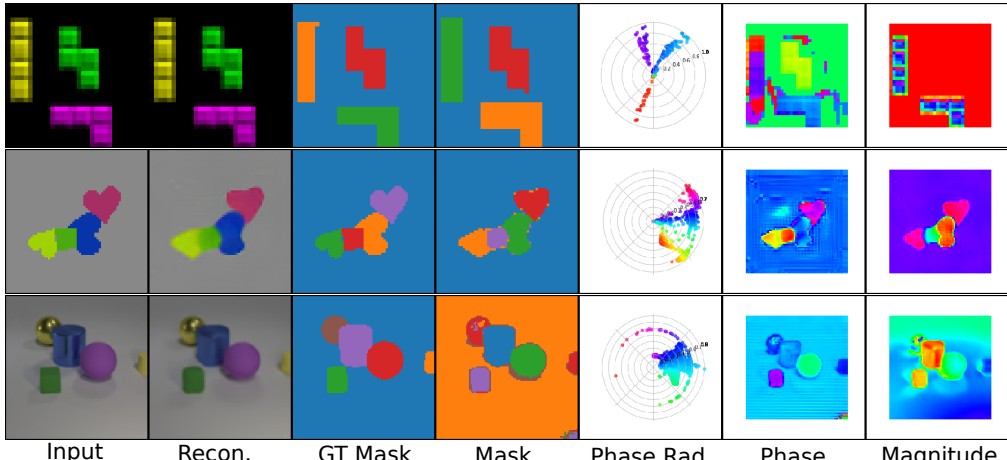

| Input | Recon. | GT Mask | Mask | Phase Rad. | Phase | Magnitude |

Figure 2: Unsupervised object discovery on `Tetrominoes`, `dSprites` and `CLEVR` with CtCAE. "Phase Rad." (col. 5) is the radial plot with the phase values from $-\pi$ to $\pi$ radians. "Phase" (col. 6), are phase values (in radians) averaged over the 3 output channels as a heatmap (colors correspond to those from "Phase Rad.") and "Magnitude" (col. 7) is the magnitude component of the outputs.

Algorithm 1, we extract 1 positive and $M - 1$ negative examples for each anchor. We denote these examples by a matrix $\mathbf{P}^k \in \mathbb{R}^{M \times d_\mathrm{f}}$ for each anchor $k \in \{1, ..., N_A\}$ arranged such that $\mathbf{P}_1^k \in \mathbb{R}^{d_\mathrm{f}}$ is the positive example and all other rows $\mathbf{P}_j^k \in \mathbb{R}^{d_\mathrm{f}}$ for $j \in \{2, ..., M\}$ are negative ones. Finally, our contrastive loss is an adaptation of the standard InfoNCE loss [44] which is defined as follows:

$$\mathcal{L}_\mathrm{ct} = \frac{1}{N_A} \sum_{k=1}^{N_A} \log \left( \frac{\exp\left(d\left(\mathbf{A}_k; \mathbf{P}_1^k\right)/\tau\right)}{\sum_{j=1}^M \exp\left(d\left(\mathbf{A}_k; \mathbf{P}_j^k\right)/\tau\right)} \right) \tag{6}$$

where $N_A$ is the number of anchors sampled for each input, $d(\mathbf{x}_k, \mathbf{x}_l)$ refers to the cosine distance between a pair of vectors $\mathbf{x}_k, \mathbf{x}_l \in \mathbb{R}^{d_f}$ and $\tau \in \mathbb{R}_{>0}$ is the softmax temperature.

We empirically observe that applying the contrastive loss on outputs of both encoder and decoder is better than applying it on only either one (Table 5). We hypothesize that this is the case because it utilizes both high-level, abstract and global features (on the encoder-side) as well as low-level and local visual cues (on the decoder-side) that better capture visual (dis)similarity between positive and negative pairs. We also observe that using magnitude components of complex outputs of both the encoder and decoder as *addresses* for mining positive and negative pairs while using the phase components of complex-valued outputs as the *features* for the contrastive loss performs the best among all the other possible alternatives (Table 13). These ablations also support our initial intuitions (described above) while designing the contrastive objective for improving *separability* in phase space.

Finally, the complete training objective function of CtCAE is:

$$\mathcal{L} = \mathcal{L}_\mathrm{mse} + \beta \cdot \mathcal{L}_\mathrm{ct} \quad ; \quad \mathcal{L}_\mathrm{mse} = ||\mathbf{x}' - \hat{\mathbf{x}}||_2^2 \quad ; \quad \hat{\mathbf{x}} = |\mathbf{y}| \tag{7}$$

where $\mathcal{L}$ defines the loss for a single input image $\mathbf{x}' \in \mathbb{R}^{h \times w \times 3}$, and $\mathcal{L}_\mathrm{mse}$ is the standard reconstruction loss used by the CAE [36]. The reconstructed image $\hat{\mathbf{x}} \in \mathbb{R}^{h \times w \times 3}$ is generated from the complex-valued outputs of the decoder $\mathbf{y} \in \mathbb{C}^{h \times w \times 3}$ by using its magnitude component. In practice, we train all models by minimizing the training loss $\mathcal{L}$ over a batch of images. The CAE baseline model and our proposed CAE++ variant are trained using only the reconstruction objective (i.e. $\beta = 0$) whereas our proposed CtCAE model is trained using the complete training objective.

# 4 Results

Here we provide our experimental results. We first describe details of the datasets, baseline models, training procedure and evaluation metrics. We then show results (always across 5 seeds) on grouping of our CtCAE model compared to the baselines (CAE and our variant CAE++), *separability* in phase space, generalization capabilities w.r.t to number of objects seen at train/test time and ablation studies for each of our design choices. Finally, we comment on the limitations of our proposed method.

Table 1: MSE and ARI scores (mean ± standard deviation across 5 seeds) for CAE, CAE++ and CtCAE models for `Tetrominoes`, `dSprites` and `CLEVR` on their respective full resolutions. For all datasets, CtCAE vastly outperforms CAE++ which in turn outperforms the CAE baseline. Results for 32x32 `dSprites` and `CLEVR` are also provided, these follow closely the scores on the full resolutions. SlotAttention results are from Emami et al. [45].

| Dataset | Model | MSE ↓ | ARI-FG ↑ | ARI-FULL ↑ |
|---|---|---|---|---|
| Tetrominoes (32x32) | CAE | 4.57e-2 ± 1.08e-3 | 0.00 ± 0.00 | 0.12 ± 0.02 |
| | CAE++ | 5.07e-5 ± 2.80e-5 | 0.78 ± 0.07 | 0.84 ± 0.01 |
| | CtCAE | 9.73e-5 ± 4.64e-5 | **0.84 ± 0.09** | **0.85 ± 0.01** |
| | SlotAttention | – | 0.99 ± 0.00 | – |
| dSprites (64x64) | CAE | 8.16e-3 ± 2.54e-5 | 0.05 ± 0.02 | 0.10 ± 0.02 |
| | CAE++ | 1.60e-3 ± 1.33e-3 | 0.51 ± 0.08 | 0.54 ± 0.14 |
| | CtCAE | 1.56e-3 ± 1.58e-4 | **0.56 ± 0.11** | **0.90 ± 0.03** |
| | SlotAttention | – | 0.91 ± 0.01 | – |
| CLEVR (96x96) | CAE | 1.50e-3 ± 4.53e-4 | 0.04 ± 0.03 | 0.18 ± 0.06 |
| | CAE++ | 2.41e-4 ± 3.45e-5 | 0.27 ± 0.13 | 0.31 ± 0.07 |
| | CtCAE | 3.39e-4 ± 3.65e-5 | **0.54 ± 0.02** | **0.68 ± 0.08** |
| | SlotAttention | – | 0.99 ± 0.01 | – |
| dSprites (32x32) | CAE | 7.24e-3 ± 8.45e-5 | 0.01 ± 0.00 | 0.05 ± 0.00 |
| | CAE++ | 8.67e-4 ± 1.92e-4 | 0.38 ± 0.05 | 0.49 ± 0.15 |
| | CtCAE | 1.10e-3 ± 2.59e-4 | **0.48 ± 0.03** | **0.68 ± 0.13** |
| CLEVR (32x32) | CAE | 1.84e-3 ± 5.68e-4 | 0.11 ± 0.07 | 0.12 ± 0.11 |
| | CAE++ | 4.04e-4 ± 4.04e-4 | 0.22 ± 0.10 | 0.30 ± 0.18 |
| | CtCAE | 9.88e-4 ± 1.42e-3 | **0.50 ± 0.05** | **0.69 ± 0.25** |

**Datasets.** We evaluate the models on three datasets from the Multi-Object datasets suite [41] namely `Tetrominoes`, `dSprites` and `CLEVR` (Figure 2) used by prior work in object-centric learning [14, 15, 45]. For `CLEVR`, we use the filtered version [45] which consists of images containing less than seven objects. For the main evaluation, we use the same image resolution as Emami et al. [45], i.e., 32x32 for `Tetrominoes`, 64x64 for `dSprites` and 96x96 for `CLEVR` (a center crop of 192x192 that is then resized to 96x96). For computational reasons, we perform all ablations and analysis on 32x32 resolution. Performance of all models are ordered in the same way on 32x32 resolution as the original resolution (see Table 1), but with significant training and evaluation speed up. In `Tetrominoes` and `dSprites` the number of training images is 60K whereas in `CLEVR` it is 50K. All three datasets have 320 test images on which we report all the evaluation metrics. For more details about the datasets and preprocessing, please refer to Appendix A.

**Models & Training Details.** We compare our CtCAE model to the state-of-the-art synchrony-based method for unsupervised object discovery (CAE [36]) as well as to our own improved version thereof (CAE++) introduced in Section 3. For more details about the encoder and decoder architecture of all models see Appendix A. We use the same architecture as CAE [36], except with increased number of convolution channels (same across all models). We train models for 50K steps on `Tetrominoes`, and 100K steps on `dSprites` and `CLEVR` with Adam optimizer [46] with a constant learning rate of 4e-4, i.e., no warmup schedules or annealing (all hyperparameter details are given in Appendix A).

**Evaluation Metrics.** We use the same evaluation protocol as prior work [14, 15, 45, 36] which compares the grouping performance of models using the Adjusted Rand Index (ARI) [47, 48]. We report two variants of the ARI score, i.e., ARI-FG and ARI-FULL consistent with Löwe et al. [36]. ARI-FG measures the ARI score only for the foreground and ARI-FULL takes into account all pixels.

**Unsupervised Object Discovery.** Table 1 shows the performance of our CAE++, CtCAE, and the baseline CAE [36] on `Tetrominoes`, `dSprites` and `CLEVR`. We first observe that the CAE baseline almost completely fails on all datasets as shown by its very low ARI-FG and ARI-FULL scores. The MSE values in Table 1 indicate that CAE even struggles to reconstruct these color

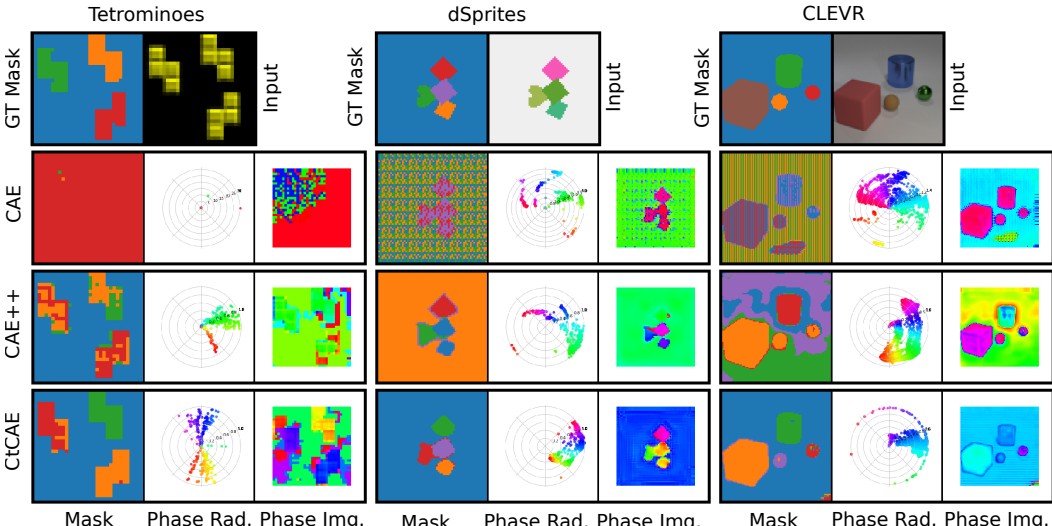

Figure 3: CAE, CAE++ and CtCAE comparison on `Tetrominoes` (columns 1-3), `dSprites` (columns 4-6) and `CLEVR` (columns 7-9). First row: ground truth masks and input images.

Table 2: Quantifying the *Separability* through inter- and intra-cluster metrics of the phase space. For the inter-cluster metric, we report both the minimum and mean across clusters.

| Dataset | Model | Inter-cluster (min) ↑ | Inter-cluster (mean) ↑ | Intra-cluster ↓ |
|---------|-------|----------------------|------------------------|-----------------|
| Tetrominoes | CAE++ | $0.14 \pm_{0.00}$ | $0.30 \pm_{0.02}$ | $0.022 \pm_{0.010}$ |
| | CtCAE | $\mathbf{0.15} \pm_{\mathbf{0.01}}$ | $\mathbf{0.31} \pm_{\mathbf{0.03}}$ | $\mathbf{0.020} \pm_{\mathbf{0.010}}$ |
| dSprites | CAE++ | $\mathbf{0.13} \pm_{\mathbf{0.05}}$ | $\mathbf{0.51} \pm_{\mathbf{0.05}}$ | $0.034 \pm_{0.007}$ |
| | CtCAE | $\mathbf{0.13} \pm_{\mathbf{0.03}}$ | $0.39 \pm_{0.10}$ | $\mathbf{0.027} \pm_{\mathbf{0.009}}$ |
| CLEVR | CAE++ | $0.10 \pm_{0.06}$ | $\mathbf{0.53} \pm_{\mathbf{0.15}}$ | $0.033 \pm_{0.013}$ |
| | CtCAE | $\mathbf{0.12} \pm_{\mathbf{0.05}}$ | $0.50 \pm_{0.12}$ | $\mathbf{0.024} \pm_{\mathbf{0.005}}$ |

images. In contrast, CAE++ achieves significantly higher ARI scores, consistently across all three datasets; this demonstrates the impact of the architectural modifications we propose. However, on the most challenging `CLEVR` dataset, CAE++ still achieves relatively low ARI scores. Its contrastive learning-augmented counterpart, CtCAE consistently outperforms CAE++ both in terms of ARI-FG and ARI-FULL metrics. Notably, CtCAE achieves more than double the ARI scores of CAE++ on the most challenging `CLEVR` dataset which highlights the benefits of our contrastive method. All these results demonstrate that CtCAE is capable of object discovery (still far from perfect) on all datasets which include color images and more than three objects per scene, unlike the exisiting state-of-the-art synchrony-based model, CAE.

**Quantitative Evaluation of *Separability*.** To gain further insights into why our contrastive method is beneficial, we quantitatively analyse the phase maps using two distance metrics: *inter-cluster* and *intra-cluster* distances. In fact, in all CAE-family of models, final object grouping is obtained through K-means clustering based on the phases at the output of the decoder; each pixel is assigned to a cluster with the corresponding centroid. *Inter*-cluster distance measures the Euclidean distance between centroids of each pair of clusters averaged over all such pairs. Larger inter-cluster distance allows for easier discriminability during clustering to obtain object assignments from phase maps. On the other hand, *intra*-cluster distance quantifies the "concentration" of points within a cluster, and is computed as the average Euclidean distance between each point in the cluster and the cluster centroid. Smaller intra-cluster distance results in an easier clustering task as the clusters are then more condensed. We compute these distance metrics on a per-image basis before averaging over all samples in the dataset. The results in Table 2 show that, the mean intra-cluster distance (last column) is smaller for CtCAE than CAE++ on two (dSprites and CLEVR) of the three datasets. Also, even though the average inter-cluster distance (fourth column) is sometimes higher for CAE++, the *minimum* inter-cluster distance (third column)—which is a more relevant metric for separability—is

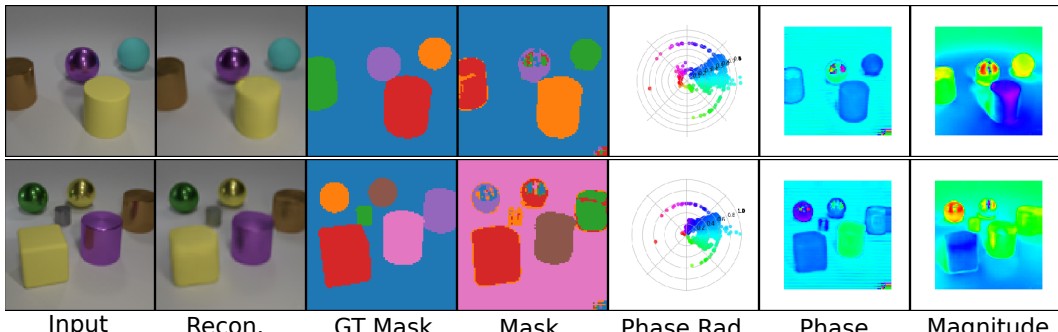

| Input | Recon. | GT Mask | Mask | Phase Rad. | Phase | Magnitude |

Figure 4: CtCAE on `CLEVR` is able to infer more than four objects, although it sometimes makes mistakes, such as specular effects or grouping together based on color (two yellow objects).

Table 3: Object storage capacity: training on the full `dSprites` dataset and evaluating separately on subsets containing images with 2, 3, 4 or 5 objects. Analogously, training on the full `CLEVR` dataset and evaluating separately on subsets containing images with 3, 4, 5 or 6 objects.

| | ARI-FG | ARI-FULL | ARI-FG | ARI-FULL | ARI-FG | ARI-FULL | ARI-FG | ARI-FULL |
|---|---|---|---|---|---|---|---|---|
| dSprites | 2 objects | | 3 objects | | 4 objects | | 5 objects | |
| CAE++ | $0.29 \pm 0.07$ | $0.58 \pm 0.15$ | $0.39 \pm 0.06$ | $0.53 \pm 0.15$ | $0.43 \pm 0.04$ | $0.45 \pm 0.13$ | $0.45 \pm 0.05$ | $0.44 \pm 0.11$ |
| CtCAE | $\mathbf{0.45} \pm \mathbf{0.11}$ | $\mathbf{0.76} \pm \mathbf{0.07}$ | $\mathbf{0.48} \pm \mathbf{0.06}$ | $\mathbf{0.72} \pm \mathbf{0.11}$ | $\mathbf{0.48} \pm \mathbf{0.04}$ | $\mathbf{0.65} \pm \mathbf{0.13}$ | $\mathbf{0.48} \pm \mathbf{0.04}$ | $\mathbf{0.61} \pm \mathbf{0.13}$ |
| CLEVR | 3 objects | | 4 objects | | 5 objects | | 6 objects | |
| CAE++ | $0.32 \pm 0.06$ | $0.35 \pm 0.29$ | $0.33 \pm 0.05$ | $0.32 \pm 0.26$ | $0.31 \pm 0.04$ | $0.26 \pm 0.20$ | $0.32 \pm 0.04$ | $0.27 \pm 0.18$ |
| CtCAE | $\mathbf{0.53} \pm \mathbf{0.07}$ | $\mathbf{0.71} \pm \mathbf{0.20}$ | $\mathbf{0.52} \pm \mathbf{0.06}$ | $\mathbf{0.70} \pm \mathbf{0.21}$ | $\mathbf{0.47} \pm \mathbf{0.07}$ | $\mathbf{0.67} \pm \mathbf{0.21}$ | $\mathbf{0.45} \pm \mathbf{0.06}$ | $\mathbf{0.68} \pm \mathbf{0.20}$ |

larger for CtCAE. This confirms that compared to CAE++, CtCAE tends to have better phase map properties for object grouping, as is originally motivated by our contrastive method.

**Object Storage Capacity.** Löwe et al. [36] note that the performance of CAE sharply decreases for images with more than 3 objects. We report the performance of CtCAE and CAE++ on subsets of the test set split by the number of objects, to measure how their grouping performance changes w.r.t. the number of objects. In `dSprites` the images contain $2, 3, 4$ or $5$ objects, in `CLEVR` $3, 4, 5$ or $6$ objects. Note that the models are trained on the entire training split of the respective datasets. Table 3 shows that both methods perform well on images containing more than 3 objects (their performance does not drop much on images with $4$ or more objects). We also observe that the CtCAE consistently maintains a significant lead over CAE in terms of ARI scores across different numbers of objects. In another set of experiments (see Appendix B), we also show that CtCAE generalizes well to more objects (e.g. $5$ or $6$) when trained only on a subset of images containing less than this number of objects.

**Ablation on Architectural Modifications.** Table 4 shows an ablation study on the proposed architectural modifications on `Tetrominoes` (for similar findings on other datasets, see Appendix B.2). We observe that the sigmoid activation on the output layer of the decoder significantly impedes learning on color datasets. A significant performance jump is also observed when replacing transposed convolution layers [36] with

Table 4: Architectural ablations on `Tetrominoes`.

| Model | ARI-FG ↑ | ARI-FULL ↑ |
|---|---|---|
| CAE | $0.00 \pm 0.00$ | $0.12 \pm 0.02$ |
| CAE-($f_{\text{out}}$ 1x1 conv) | $0.00 \pm 0.00$ | $0.00 \pm 0.00$ |
| CAE-($f_{\text{out}}$ sigmoid) | $0.12 \pm 0.12$ | $0.35 \pm 0.36$ |
| CAE-transp.+upsamp. | $0.10 \pm 0.21$ | $0.10 \pm 0.22$ |
| CAE++ | $0.78 \pm 0.07$ | $0.84 \pm 0.01$ |
| CtCAE | $0.84 \pm 0.09$ | $0.85 \pm 0.01$ |

convolution and upsample layers. By applying all these modifications, we obtain our CAE++ model that results in significantly better ARI scores on all datasets, therefore supporting our design choices.

**Ablation on Feature Layers to Contrast.** In CtCAE we contrast feature vectors both at the output of the encoder and the output of the decoder. Table 5 justifies this choice; this default setting (Enc+Dec) outperforms both other options where we apply the constrastive loss either only in the

encoder or the decoder output. We hypothesize that this is because these two contrastive strategies are complementary: one uses low-level cues (dec) and the other high-level abstract features (enc).

**Qualitative Evaluation of Grouping.** Finally, we conduct some qualitative analyses of both successful and failed grouping modes shown by CAE++ and CtCAE models through visual inspection of representative samples. In Figure 3, `Tetrominoes` (columns 1-2), we observe that CtCAE (row 4) exhibits better grouping on scenes with multiple objects of the same color than CAE++ (row 3). This is reflected in the radial phase

Table 5: Contrastive loss ablation for CtCAE on `CLEVR`.

| Model | ARI-FG ↑ | ARI-FULL ↑ |
|---|---|---|
| Enc-only | $0.21 \pm 0.11$ | $0.29 \pm 0.15$ |
| Dec-only | $0.38 \pm 0.17$ | $0.69 \pm 0.18$ |
| Enc+Dec | $0.50 \pm 0.05$ | $0.69 \pm 0.25$ |

plots (column 2) which show better *separability* for CtCAE than CAE++. Further, on `dSprites` (rows 3-4, columns 4-5) and `CLEVR` (rows 3-4, columns 7-8), CtCAE handles the increased number of objects more gracefully while CAE++ struggles and groups several of them together. For the failure cases, Figure 4 shows an example where CtCAE still has some difficulties in segregating objects of the same color (row 2, yellow cube and ball) (also observed sometimes on `dSprites`, see Figure 14). Further, we observe how the *specular highlights* on metallic objects (purple ball in row 1 and yellow ball in row 2) form a separate sub-part from the object (additional grouping examples in Appendix C).

**Discussion on the Evaluation Protocol.** The reviewers raised some concerns about the validity of our evaluation protocol[3]. The main point of contention was that the thresholding applied before clustering the phase values may lead to a trivial separation of objects in RGB images based on their color. While it is true that in our case certain color information may contribute to the separation process, a trivial separation of objects based on solely their color cannot happen since CtCAE largely outperforms CAE++ (see Table 1) despite both having near perfect reconstruction losses and with the same evaluation protocol. This indicates that this potential issue only has a marginal effect in practice in our case and separation in phase space learned by the model is still crucial. In fact, "pure" RGB color tones (which allow such trivial separation) rarely occur in the datasets (`dSprites` and `CLEVR`) used here. The percentage of pixels with a magnitude less than 0.1 are $0.44\%$, $0.88\%$ and $9.25\%$ in `CLEVR`, `dSprites` and `Tetrominoes` respectively. While $9.25\%$ in `Tetrominoes` is not marginal, we observe that this does not pose an issue in practice, as many examples in `Tetrominoes` where two or three blocks with the same color are separated by CAE++/CtCAE (e.g. three yellow blocks in Figure 3 or two magenta blocks in Figure 8).

To support our argument that using a threshold of 0.1 has no effect on the findings and conclusions, we conduct additional evaluations with a threshold of 0 and no threshold at all. From Table 9 (see Appendix B) we can see that using a threshold of 0 hardly change the ARI scores at all or in some cases it even improves the scores slightly. Using no threshold results in marginally improved ARI scores on `CLEVR`, near-identical scores on `dSprites` but worse scores on `Tetrominoes` (CtCAE still widely outperforms CAE++). However, this is not due to a drawback in the evaluation or our model. Instead, it stems from an inherent difficulty with estimating the phase of a zero-magnitude complex number. The evaluation clusters pixels based on their phase values, and if a complex number has a magnitude of exactly zero (or very small, e.g., order of $1e^{-5}$), the phase estimation is ill-defined and will inherently be random as was also previously noted by Löwe et al. [36]. In fact, filtering those pixels in the evaluation is crucial in their work [36] as they largely used datasets with pure black background (exact zero pixel value). Evaluation of discrete object assignments from continuous phase outputs in a model-agnostic way remains an open research question.

## 5   Related Work

**Slot-based binding.** A wide range of unsupervised models have been introduced to perform perceptual grouping summarized well by Greff et al. [30]. They categorize models based on the segregation (routing) mechanism used to break the symmetry in representations and infer latent representations (i.e. slots). Models that use "instance slots" cast the routing problem as inference in a mixture model whose solution is given by amortized variational inference [8, 14], Expectation-Maximization [10] or other approximations (Soft K-means) thereof [15]. While others [9, 12, 49] that use "sequential slots" solve the routing problem by imposing an ordering across time. These

---

[3]Discussion thread: `https://openreview.net/forum?id=nF6X3uOFaA&noteId=5BeEERfCvI`

models use recurrent neural networks and an attention mechanism to route information about a different object into the same slot at every timestep. Some models [13, 17] combine the above strategies and use recurrent attention only for routing but not for inferring slots. Other models break the representational symmetry based on spatial coordinates [50, 16] ("spatial slots") or based on specific object types [51] ("category slots"). All these models still maintain the "separation" of representations only at one latent layer (slot-level) but continue to "entangle" them at other layers.

**Synchrony-based binding.** Synchrony-based models use complex-valued activations to implement binding by relying on their constructive or destructive phase interference phenomena. This class of models have been sparsely explored with only few prior works that implement this conceptual design for object binding [38–40, 36]. These methods differ based on whether they employ both complex-valued weights and activations [38, 39] or complex-valued activations with real-valued weights and a gating mechanism [40, 36]. They also differ in their reliance on explicit supervision for grouping [38] or not [39, 40, 36]. Synchrony-based models in contrast to slot-based ones maintain the "separation" of representations throughout the network in the phase components of their complex-valued activations. However, none of these prior methods can group objects in color images with up to 6 objects or visual realism of multi-object benchmarks in a fully unsupervised manner unlike ours. Concurrent work [52] extends CAE by introducing new feature dimensions ("rotating features"; RF) to the complex-valued activations. However, RF cannot be directly compared to CtCAE as it relies on either depth masks to get instance grouping on simple colored `Shapes` or features from powerful vision backbones (DINO [53]) to get largely semantic grouping on real-world images such as multiple instances of cows/trains or bicycles in their Figures 2 and 20 respectively.

**Binding in the brain.** The temporal correlation hypothesis posits that the mammalian brain binds together information emitted from groups of neurons that fire synchronously. According to this theory [54, 55], biological neurons transmit information in two ways through their spikes. The spike amplitude indicates the strength of presence of a feature while relative time between spikes indicates which neuronal responses need to bound together during further processing. It also suggests candidate rhythmic cycles in the brain such as Gamma that could play this role in binding [56, 57]. Synchrony-based models functionally implement the same coding scheme [58–60] using complex-valued activations where the relative phase plays the role of relative time between spikes. This abstracts away all aspects of the spiking neuron model to allow easier reproduction on digital hardware.

**Contrastive learning with object-centric models.** Contrastive learning for object-centric representations has not been extensively explored, with a few notable exceptions. The first method [61] works only on toy images of up to 3 objects on a black background while the second [62] shows results on more complex data, but requires complex hand-crafted data augmentation techniques to contrast samples across a batch. Our method samples positive and negative pairs *within* a single image and does not require any data augmentation. Most importantly, unlike ours, these models still use slots and attention-based routing, and thereby inherit all of its conceptual limitations. Lastly, ODIN [63] alternately refines the segmentation masks and representation of objects using two networks that are jointly optimized by a contrastive objective that maximizes the similarity between different views of the same object, while minimizing the similarity between different objects. To obtain the segmentation masks for objects, they simply spatially group the features using K-means. Their method relies on careful data augmentation techniques such as local or global crops, viewpoint changes, color perturbations etc. which are applied to one object.

## 6 Conclusion

We propose several architectural improvements and a novel contrastive learning method to address limitations of the current state-of-the-art synchrony-based model for object binding, the complex-valued autoencoder (CAE [36]). Our improved architecture, CAE++, is the first synchrony-based model capable of dealing with color images (e.g., `Tetrominoes`). Our contrastive learning method further boosts CAE++ by improving its phase separation process. The resulting model, CtCAE, largely outperforms CAE++ on the rather challenging `CLEVR` and `dSprites` datasets. Admittedly, our synchrony-based models still lag behind the state-of-the-art *slot*-based models [15, 18], but this is to be expected, as research on modern synchrony-based models is still in its infancy. We hope our work will inspire the community to invest greater effort into such promising models.

**Acknowledgments.** We thank Sindy Löwe and Michael C. Mozer for insightful discussions and valuable feedback. This research was funded by Swiss National Science Foundation grant: 200021_192356, project NEUSYM and the ERC Advanced grant no: 742870, AlgoRNN. This work was also supported by a grant from the Swiss National Supercomputing Centre (CSCS) under project ID s1205 and d123. We also thank NVIDIA Corporation for donating DGX machines as part of the Pioneers of AI Research Award.

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

## A   Experimental Details

**Datasets.**   We evaluate all models (i.e., CAE, CAE++ and CtCAE) on a subset of Multi-Object dataset [41] suite. We use three datasets: `Tetrominoes` which consists of colored tetris blocks on a black background, `dSprites` with colored sprites of various shapes like heart, square, oval, etc., on a grayscale background, and lastly, `CLEVR`, a dataset from a synthetic 3D environment. For `CLEVR`, we use the filtered version [45] which consists of images containing less than seven objects sometimes referred to as `CLEVR6` as in Locatello et al. [15]. We normalize all input RGB images to have pixel values in the range $[0, 1]$ consistent with prior work [36].

**Models.**   Table 6 shows the architecture specifications such as number of layers, kernel sizes, stride lengths, number of filter channels, normalization layers, activations, etc., for the convolutional neural networks used by the Encoder and Decoder modules in CAE, CAE++ and CtCAE.

Table 6: Encoder and Decoder architecture specifications for CAE, CAE++ and CtCAE models.

| **Encoder** |
| --- |
| $3 \times 3$ conv, 128 channels, stride 2, ReLU, BatchNorm |
| $3 \times 3$ conv, 128 channels, stride 1, ReLU, BatchNorm |
| $3 \times 3$ conv, 256 channels, stride 2, ReLU, BatchNorm |
| $3 \times 3$ conv, 256 channels, stride 1, ReLU, BatchNorm |
| $3 \times 3$ conv, 256 channels, stride 2, ReLU, BatchNorm |
| *For 64×64 and 96×96 inputs, 2 additional encoder layers:* |
| $3 \times 3$ conv, 256 channels, stride 1, ReLU, BatchNorm |
| $3 \times 3$ conv, 256 channels, stride 2, ReLU, BatchNorm |
| **Linear Layer** |
| Flatten |
| Linear, 256 ReLU units, LayerNorm |
| **Decoder** |
| *For 64×64 and 96×96 inputs, 2 additional decoder layers:* |
| Bilinear upsample x2 |
| $3 \times 3$ conv, 256 channels, stride 1, ReLU, BatchNorm |
| $3 \times 3$ conv, 256 channels, stride 1, ReLU, BatchNorm |
| Bilinear upsample x2 |
| $3 \times 3$ conv, 256 channels, stride 1, ReLU, BatchNorm |
| $3 \times 3$ conv, 256 channels, stride 1, ReLU, BatchNorm |
| Bilinear upsample x2 |
| $3 \times 3$ conv, 256 channels, stride 1, ReLU, BatchNorm |
| $3 \times 3$ conv, 128 channels, stride 1, ReLU, BatchNorm |
| Bilinear upsample x2 |
| $3 \times 3$ conv, 128 channels, stride 1, ReLU, BatchNorm |
| **Output Layer (CAE only)** |
| $1 \times 1$ conv, 3 channels, stride 1, sigmoid |

**Training Details.**   Table 7 and Table 8 show the hyperparameter configurations used to train CAE/CAE++ and to contrastively train the CtCAE model(s) respectively.

**Computational Efficiency.**   We report the training and inference time (wall-clock) for our models across the various image resolutions of the 3 multi-object datasets. First, we find that inference time(s) is similar for all models, and it only depends on the image resolution and number of ground truth clusters in the input image. Inference for the test set containing 320 images of 32x32 resolution takes 25 seconds, for 64x64 images it takes 65 seconds and for the 96x96 images it takes 105 seconds. Training time(s) on the other hand differs both across models and image resolutions. We report all

Table 7: General training hyperparameters.

| Hyperparameter | Tetrominoes | dSprites | CLEVR |
|---|---|---|---|
| Training Steps | 50'000 | 100'000 | 100'000 |
| Batch size | 64 | 64 | 64 |
| Learning rate | 4e-4 | 4e-4 | 4e-4 |

Table 8: Contrastive learning hyperparameters for the CtCAE model.

| Common Hyperparameters | | | |
|---|---|---|---|
| Loss coefficient (in total loss sum) | 1e-4 | | |
| Temperature | 0.05 | | |
| Contrastive Learning Addresses | Magnitude | | |
| Contrastive Learning Features | Phase | | |
| Encoder Hyperparameters | 32x32 | 64x64 | 96x96 |
| Number of anchors | 4 | 4 | 6 |
| Number of positive pairs | 1 | 1 | 1 |
| Top-K to select positive pairs from | 1 | 1 | 5 |
| Number of negative pairs | 2 | 2 | 6 |
| Bottom-M to select negative pairs from | 2 | 2 | 18 |
| Decoder Hyperparameters | 32x32 | 64x64 | 96x96 |
| Patch Size | 1 | 2 | 3 |
| Number of anchors | 100 | 100 | 100 |
| Number of positive pairs | 1 | 1 | 1 |
| Top-K to select positive pairs from | 5 | 5 | 5 |
| Number of negative pairs | 100 | 100 | 100 |
| Bottom-M to select negative pairs from | 500 | 500 | 500 |

training time(s) using a single Nvidia V100 GPU. CAE and CAE++ have similar training times: for 32x32 images training for 100k steps takes 3.2 hours, for 64x64 images it takes 4.8 hours, and for 96x96 images it takes 7 hours. CtCAE on the other hand takes 5 hours, 9.2 hours and 10.2 hours to train on dataset of 32x32, 64x64 and 96x96 images respectively. To reproduce all the results/tables (mean and std-dev across 5 seeds) reported in this work we estimate the compute requirement to be 840 GPU hours in total. Further, we estimate that the total compute used in this project is approximately 5-10x more, including experiments with preliminary prototypes.

**Object Assignments from Phase Maps.** We describe the process of extracting discrete-valued object assignments from continuous-valued phase components of decoder outputs $\mathbf{y} \in \mathbb{C}^{h \times w \times 3}$. First, the phase components of decoder outputs are mapped onto a unit circle and those phase values are masked out whose magnitudes are below the threshold value of 0.1. After this normalization and filtering step, the resultant phase values are converted from polar to Cartesian form on a per-channel basis. Finally, we apply K-means clustering where the number of clusters $K$ parameter is retrieved from the ground-truth segmentation mask for each image. This extraction methodology of object assignments from output phase maps is consistent with Löwe et al. [36].

# B   Additional results

We show some additional experimental results: generalization capabilities of CtCAE w.r.t. the number of objects, and ablation studies on various design choices made in CAE++ and CtCAE models.

Table 9: Grouping results for CAE++ and CtCAE models with different threshold values applied to post-process the continuous output phase maps.

| Dataset | Model | Threshold=0.1 | | Threshold=0.0 | | No threshold | |
|---|---|---|---|---|---|---|---|
| | | ARI-FG ↑ | ARI-FULL ↑ | ARI-FG ↑ | ARI-FULL ↑ | ARI-FG ↑ | ARI-FULL ↑ |
| Tetrominoes | CAE++ | $0.78 \pm 0.07$ | $0.84 \pm 0.01$ | $0.77 \pm 0.07$ | $0.79 \pm 0.02$ | $0.54 \pm 0.09$ | $0.21 \pm 0.05$ |
| | CtCAE | $0.84 \pm 0.09$ | $0.85 \pm 0.01$ | $0.86 \pm 0.05$ | $0.82 \pm 0.01$ | $0.67 \pm 0.11$ | $0.26 \pm 0.09$ |
| dSprites | CAE++ | $0.38 \pm 0.05$ | $0.49 \pm 0.15$ | $0.38 \pm 0.05$ | $0.49 \pm 0.12$ | $0.37 \pm 0.05$ | $0.49 \pm 0.12$ |
| | CtCAE | $0.48 \pm 0.03$ | $0.68 \pm 0.13$ | $0.46 \pm 0.07$ | $0.69 \pm 0.10$ | $0.47 \pm 0.06$ | $0.69 \pm 0.10$ |
| CLEVR | CAE++ | $0.22 \pm 0.10$ | $0.30 \pm 0.18$ | $0.33 \pm 0.04$ | $0.32 \pm 0.25$ | $0.34 \pm 0.04$ | $0.32 \pm 0.25$ |
| | CtCAE | $0.50 \pm 0.05$ | $0.69 \pm 0.25$ | $0.52 \pm 0.05$ | $0.69 \pm 0.20$ | $0.52 \pm 0.06$ | $0.72 \pm 0.21$ |

Table 10: Generalization evaluation on the CLEVR dataset. Training only on a subset, but evaluating on all possible subsets containing images with 3, 4, 5 or 6 objects.

| Evaluation | ARI-FG ↑ | ARI-FULL ↑ |
|---|---|---|
| Training Subset: 4 or less objects | | |
| 3 objects | $0.39 \pm 0.20$ | $0.53 \pm 0.30$ |
| 4 objects | $\mathbf{0.41} \pm \mathbf{0.18}$ | $\mathbf{0.55} \pm \mathbf{0.29}$ |
| 5 objects | $0.38 \pm 0.15$ | $0.54 \pm 0.28$ |
| 6 objects | $0.38 \pm 0.13$ | $0.55 \pm 0.24$ |
| All images | $0.39 \pm 0.04$ | $0.54 \pm 0.28$ |
| Training Subset: 5 or less objects | | |
| 3 objects | $0.49 \pm 0.04$ | $\mathbf{0.69} \pm \mathbf{0.25}$ |
| 4 objects | $\mathbf{0.50} \pm \mathbf{0.03}$ | $0.66 \pm 0.24$ |
| 5 objects | $0.46 \pm 0.02$ | $0.65 \pm 0.24$ |
| 6 objects | $0.45 \pm 0.03$ | $0.64 \pm 0.23$ |
| All images | $0.48 \pm 0.02$ | $0.66 \pm 0.24$ |

## B.1 Generalization to Higher Number of Objects

Here we evaluate generalization capabilities of CtCAE by training only on a subset of images containing less than a certain number of objects, on CLEVR. We have two cases—training on subsets with either up to 4 or up to 5 objects while the original trainig split contain between 3 and 6 objects. The results in Table 10 show that CtCAE generalizes well. The performance drops only marginally when trained on up to 4 objects and tested on 5 and 6 objects, or when trained on up to 5 objects and tested on 6 objects. We also observe that training on 5 or less objects also consistently improves ARI scores for every subset of less than 5 objects. We suspect that the reason for this, apart from having more training data, is that the network has more pressure to separate objects in the phase space when observing more objects, which in turn also helps for images with a smaller number of objects.

## B.2 Architecture Modifications

Table 11 shows the results from the ablation study that measures the effect of applying each of our proposed architectural modifications cumulatively to finally result in the CAE++ model. Across all datasets, we consistently observe that, starting from the vanilla CAE baseline which completely fails, grouping performance gradually improves as more components of the CAE++ model is added. Lastly, our proposed contrastive objective used to train CtCAE further improves CAE++ across all 3 datasets.

## B.3 Contrastive Learning Ablations

Table 12 shows an ablation study for the choice of layer(s) to which we apply our contrastive learning method. Please note that all variants here always use magnitude components of complex-valued activations as *addresses* and phase components as *features* to contrast. We observe that the variant

Table 11: Grouping metrics achieved by various model variants from our proposed architectural modifications and resulting finally in the CAE++ model. Extends the results of Table 4 to all 3 multi-object datasets.

| Dataset | Model | ARI-FG ↑ | ARI-FULL ↑ |
|---------|-------|----------|------------|
| Tetrominoes | CAE | $0.00 \pm 0.00$ | $0.12 \pm 0.02$ |
| | CAE-($f_{out}$ 1x1 conv) | $0.00 \pm 0.00$ | $0.00 \pm 0.00$ |
| | CAE-($f_{out}$ sigmoid) | $0.12 \pm 0.12$ | $0.35 \pm 0.36$ |
| | CAE-transp.+upsamp. | $0.10 \pm 0.21$ | $0.10 \pm 0.22$ |
| | CAE++ (above combined) | $0.78 \pm 0.07$ | $0.84 \pm 0.01$ |
| | CtCAE | $\mathbf{0.84} \pm \mathbf{0.09}$ | $\mathbf{0.85} \pm \mathbf{0.01}$ |
| dSprites | CAE | $0.02 \pm 0.00$ | $0.07 \pm 0.01$ |
| | CAE-($f_{out}$ 1x1 conv) | $0.06 \pm 0.01$ | $0.07 \pm 0.07$ |
| | CAE-($f_{out}$ sigmoid) | $0.02 \pm 0.01$ | $0.07 \pm 0.01$ |
| | CAE-transp.+upsamp. | $0.19 \pm 0.02$ | $0.10 \pm 0.04$ |
| | CAE++ (above combined) | $0.38 \pm 0.05$ | $0.49 \pm 0.15$ |
| | CtCAE | $\mathbf{0.48} \pm \mathbf{0.03}$ | $\mathbf{0.68} \pm \mathbf{0.13}$ |
| CLEVR | CAE | $0.09 \pm 0.05$ | $0.08 \pm 0.06$ |
| | CAE-($f_{out}$ 1x1 conv) | $0.05 \pm 0.01$ | $0.06 \pm 0.01$ |
| | CAE-($f_{out}$ sigmoid) | $0.06 \pm 0.02$ | $0.01 \pm 0.02$ |
| | CAE-transp.+upsamp. | $0.19 \pm 0.07$ | $0.10 \pm 0.11$ |
| | CAE++ (above combined) | $0.22 \pm 0.10$ | $0.30 \pm 0.18$ |
| | CtCAE | $\mathbf{0.50} \pm \mathbf{0.05}$ | $\mathbf{0.69} \pm \mathbf{0.25}$ |

which applies the contrastive objective to both the outputs of the encoder and decoder ('enc+dec') outperforms the others which apply it only to either one ('enc-only' or 'dec-only'). This behavior is consistent across Tetrominoes, dSprites and CLEVR. We hypothesize that this is because these two contrastive strategies are complementary: one is on low-level cues (decoder) and the other one based on high-level abstract features (encoder).

Table 13 shows an ablation study for the choice of using magnitude or phase as *addresses* (and the other one as *features*) in our contrastive learning method. As we apply contrastive learning to both the decoder and encoder outputs, we can make this decision independently for each output (resulting in four possible combinations). We observe that the variant which uses magnitude for both the encoder and decoder ('mg+mg') outperforms other variants across all 3 multi-object datasets. These ablations support our initial intuitions when designing our contrastive objective.

Table 12: Grouping metrics achieved by CtCAE model variants that apply the contrastive loss on output features from only the encoder (enc-only) or only the decoder (dec-only) or both (enc+dec) for all 3 multi-object datasets. Extends the results of Table 5 to all 3 multi-object datasets.

| Dataset | Model | ARI-FG ↑ | ARI-FULL ↑ |
|---------|-------|----------|------------|
| Tetrominoes | CtCAE (enc-only) | $0.81 \pm 0.06$ | $0.85 \pm 0.01$ |
| | CtCAE (dec-only) | $0.74 \pm 0.04$ | $\mathbf{0.86} \pm \mathbf{0.00}$ |
| | CtCAE (enc+dec) | $\mathbf{0.84} \pm \mathbf{0.09}$ | $0.85 \pm 0.01$ |
| dSprites | CtCAE (enc-only) | $0.40 \pm 0.06$ | $0.58 \pm 0.05$ |
| | CtCAE (dec-only) | $0.48 \pm 0.05$ | $\mathbf{0.72} \pm \mathbf{0.07}$ |
| | CtCAE (enc+dec) | $\mathbf{0.48} \pm \mathbf{0.03}$ | $0.68 \pm 0.13$ |
| CLEVR | CtCAE (enc-only) | $0.21 \pm 0.11$ | $0.29 \pm 0.15$ |
| | CtCAE (dec-only) | $0.38 \pm 0.17$ | $0.69 \pm 0.18$ |
| | CtCAE (enc+dec) | $\mathbf{0.50} \pm \mathbf{0.05}$ | $\mathbf{0.69} \pm \mathbf{0.25}$ |

Table 13: Grouping metrics on all 3 multi-object datasets achieved by CtCAE model variants that use either magnitude or phase components of the encoder/decoder outputs as the *addresses* for contrastive learning. For example, 'mg+ph' means that magnitude components used as *addresses* of the encoder outputs and phase components used as *addresses* of the decoder outputs (conversely, the phase components of the encoder outputs are used as *features* and the magnitude components of the decoder outputs are used as *features*).

| Dataset | Model | ARI-FG ↑ | ARI-FULL ↑ |
|---|---|---|---|
| Tetrominoes | CtCAE w/ mg+mg | **0.84** ± **0.09** | **0.85** ± **0.01** |
| | CtCAE w/ ph+ph | 0.85 ± 0.05 | 0.84 ± 0.01 |
| | CtCAE w/ mg+ph | 0.86 ± 0.03 | 0.85 ± 0.02 |
| | CtCAE w/ ph+mg | 0.77 ± 0.04 | 0.85 ± 0.01 |
| dSprites | CtCAE w/ mg+mg | **0.48** ± **0.03** | **0.68** ± **0.13** |
| | CtCAE w/ ph+ph | 0.38 ± 0.08 | 0.42 ± 0.16 |
| | CtCAE w/ mg+ph | 0.36 ± 0.07 | 0.57 ± 0.13 |
| | CtCAE w/ ph+mg | 0.45 ± 0.05 | 0.67 ± 0.18 |
| CLEVR | CtCAE w/ mg+mg | **0.50** ± **0.05** | **0.69** ± **0.25** |
| | CtCAE w/ ph+ph | 0.22 ± 0.06 | 0.22 ± 0.09 |
| | CtCAE w/ mg+ph | 0.18 ± 0.08 | 0.28 ± 0.15 |
| | CtCAE w/ ph+mg | 0.40 ± 0.15 | 0.65 ± 0.19 |

## C  Additional Visualizations

We show additional visualization samples of groupings from CAE, CAE++ and CtCAE on `Tetrominoes`, `dSprites` and `CLEVR`, and qualitatively highlight both grouping failures and successes of CAE, CAE++ and CtCAE.

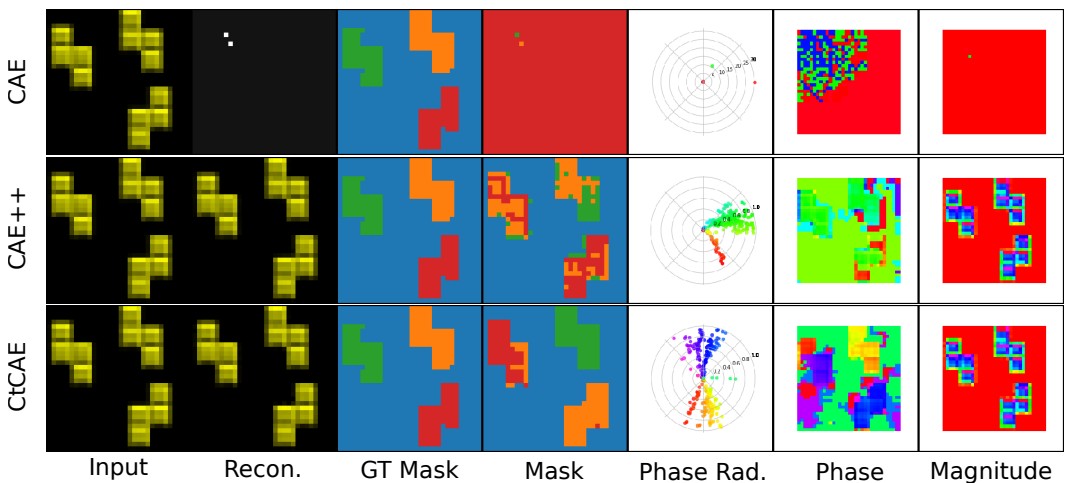

Figure 5: Visualizations for CAE (row 1), CAE++ (row 2) and CtCAE (row 3) on the `Tetrominoes` dataset example from Figure 3. We see that CtCAE achieves near perfect grouping (column 4) and better *separability* in phase space (column 5) compared to our improved CAE++ variant which shows significant grouping interference (parts of all three objects modelled by same group, e.g., orange cluster). We also observe that the baseline CAE completely fails to reconstruct or group the image.

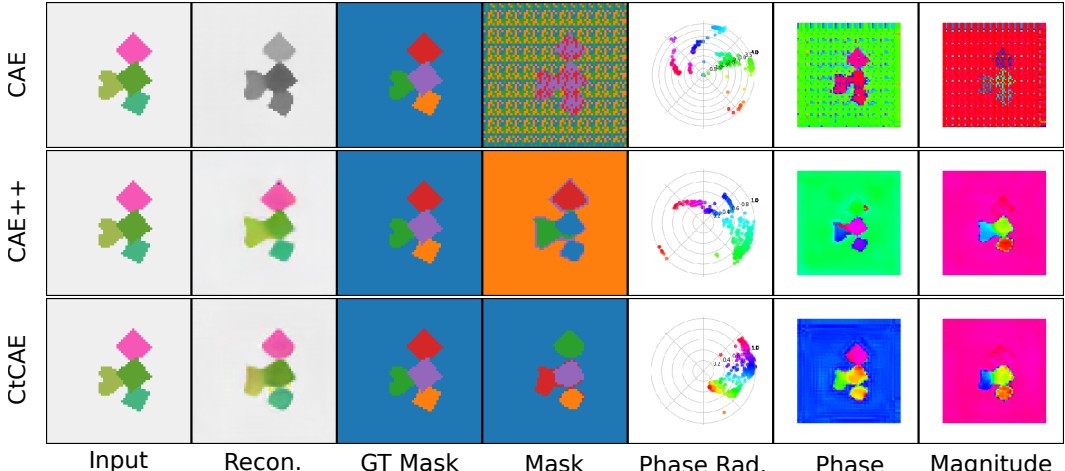

Figure 6: Visualizations for CAE (row 1), CAE++ (row 2) and CtCAE (row 3) on the dSprites dataset example from Figure 3. We see that CtCAE successfully separates the 4 scene objects into 4 clusters while CAE++ mistakenly groups 2 scene objects into 1 cluster, i.e., the blue cluster (column 4). Again, the baseline CAE fails to reconstruct (no color) or group the image.

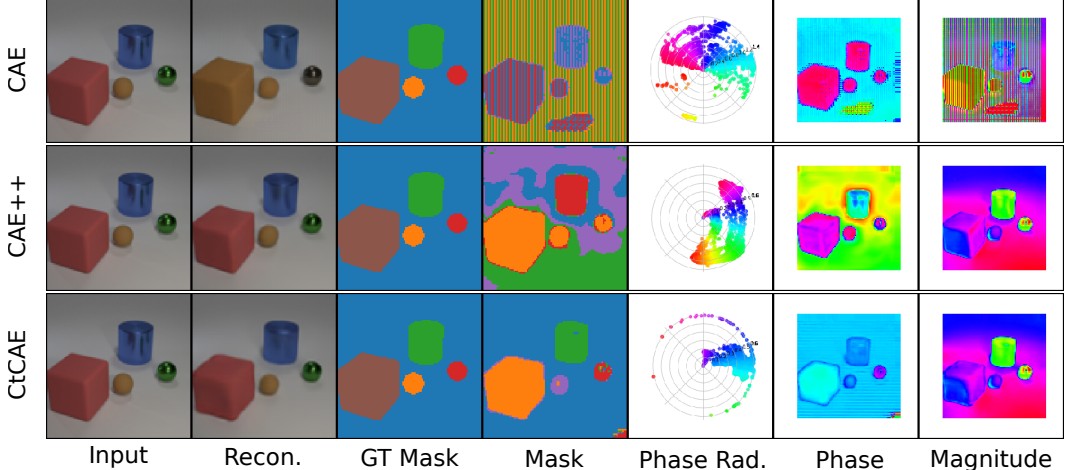

Figure 7: Visualizations for CAE (row 1), CAE++ (row 2) and CtCAE (row 3) on the CLEVR dataset example from Figure 3. We see that the CtCAE in this example is able to separate the 4 scene objects into 4 distinct clusters while our improved CAE++ fails (column 4). This good separation in phase space is reflected in the continuous phase maps (column 6) where the pixel colors in the phase maps are representative of their values/differences when comparing CtCAE and CAE++. Finally, we see that although the CAE is able to reasonably reconstruct the image it shows very poor grouping of the scene objects.

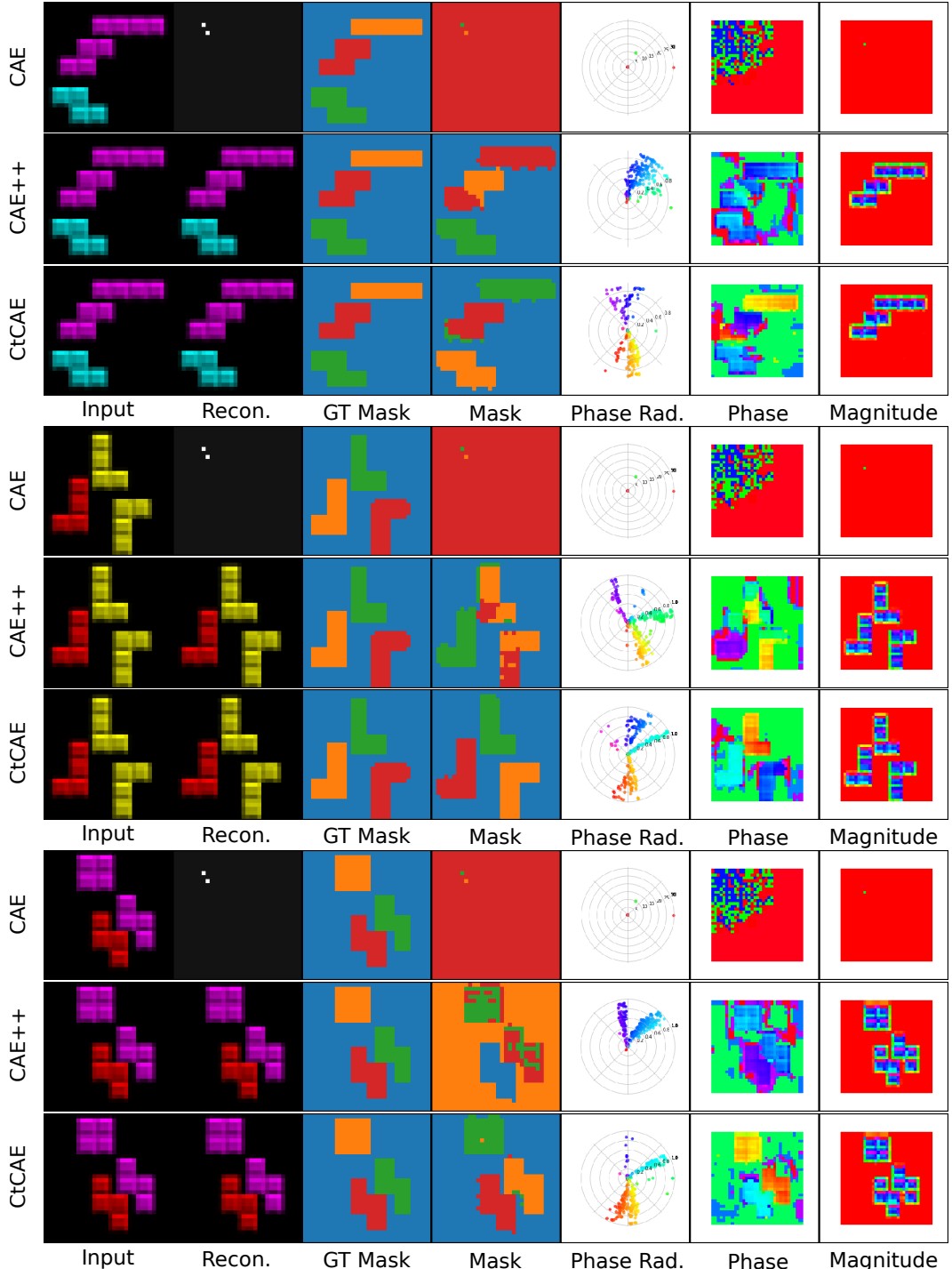

Figure 8: Visualization on grouping performance for CAE, CAE++ and CtCAE on images containing same colored objects on Tetrominoes. CAE struggles to even reconstruct the images, whereas CAE++ often groups objects with the same colour together. CtCAE on the other hand has no problem separating objects of the same colour on Tetrominoes.

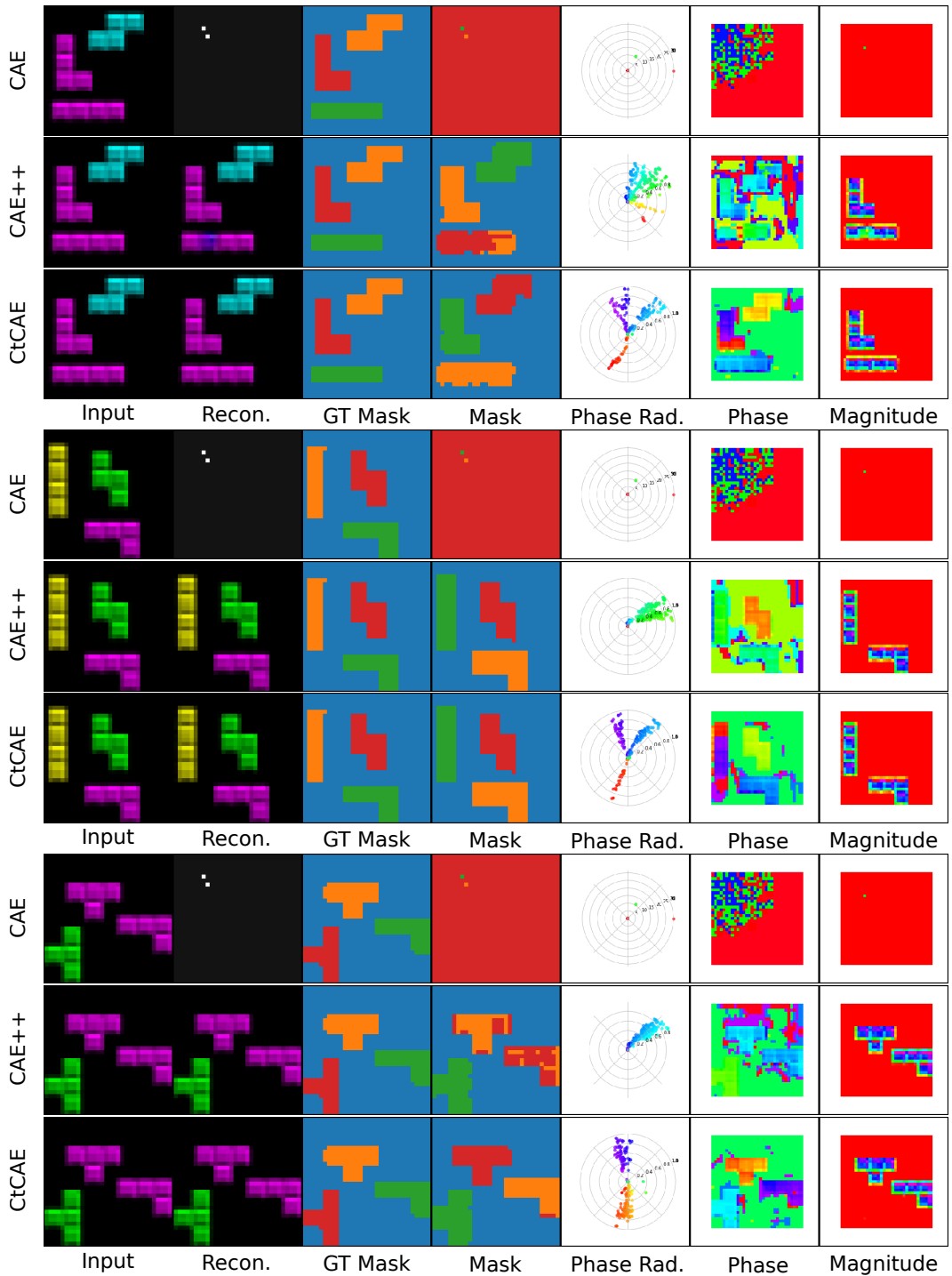

Figure 9: Visualization on grouping performance for CAE, CAE++ and CtCAE on `Tetrominoes`. CtCAE shows a larger spread in phase values for different clusters (column 5) compared to CAE. We hypothesize that this facilitates CtCAE to perform better grouping in the scenario of multiple object instances with the same color.

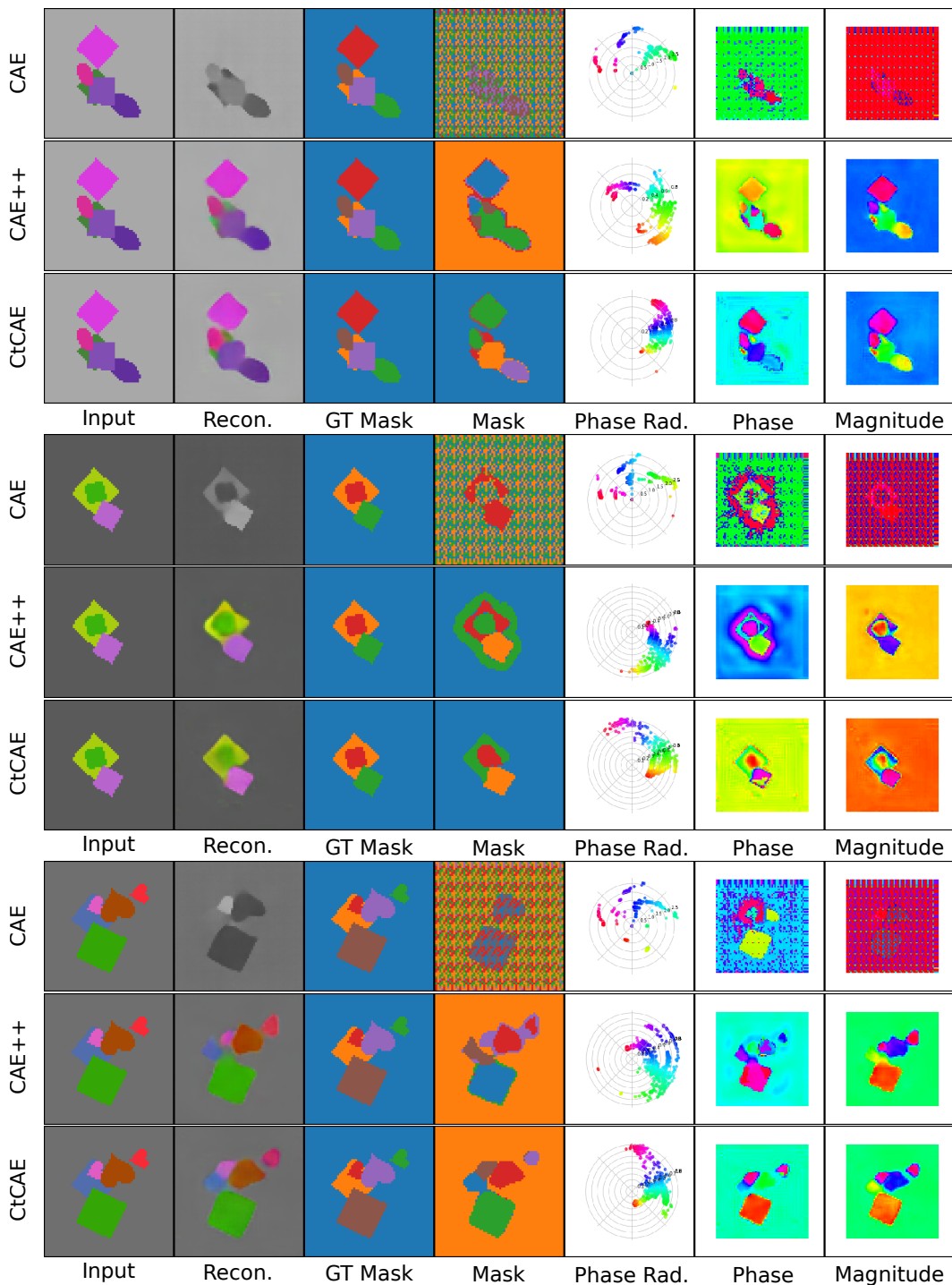

Figure 10: Visualization on grouping performance for CAE, CAE++ and CtCAE on images containing similar colored objects on dSprites. CAE has very poor grouping capability, whereas CAE++ often groups objects with the same colour together For example, purple ellipse and square in the first example (top panel) or red and brown hearts grouped together in the last example (bottom panel) are grouped together. CtCAE on the other hand has significantly better performance, grouping correctly objects with the same (or similar) colors, except in the first example (top panel) where it groups the pink square and ellipse together.

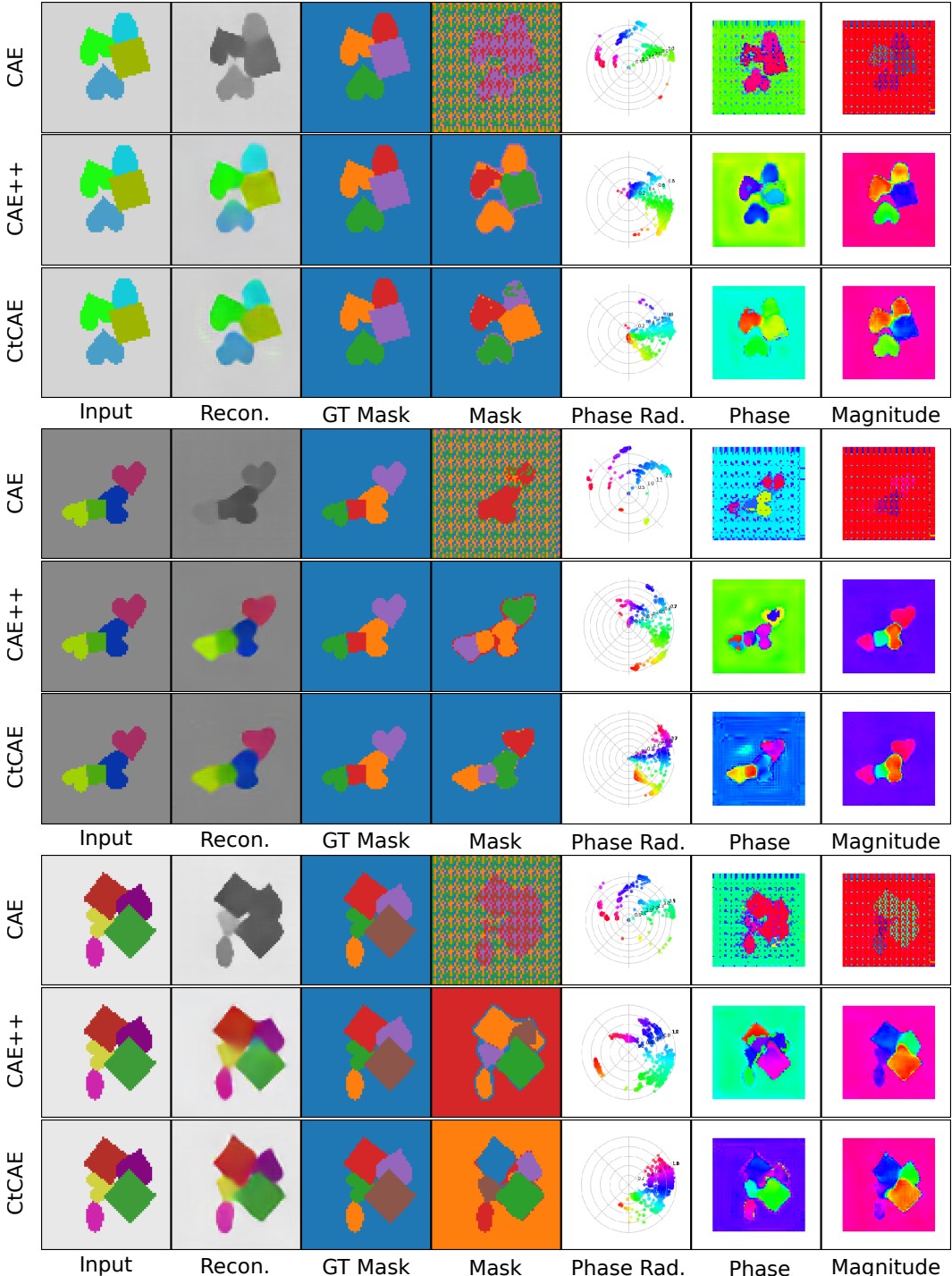

Figure 11: Performance for CAE, CAE++ and CtCAE on images containing many objects on `dSprites`. CAE shows poor grouping performance, as noted previously. CtCAE has no issues grouping up to 5 objects in the same image, whereas CAE++ struggles, often grouping multiple objects (typically of same or similar colors) together into the same cluster.

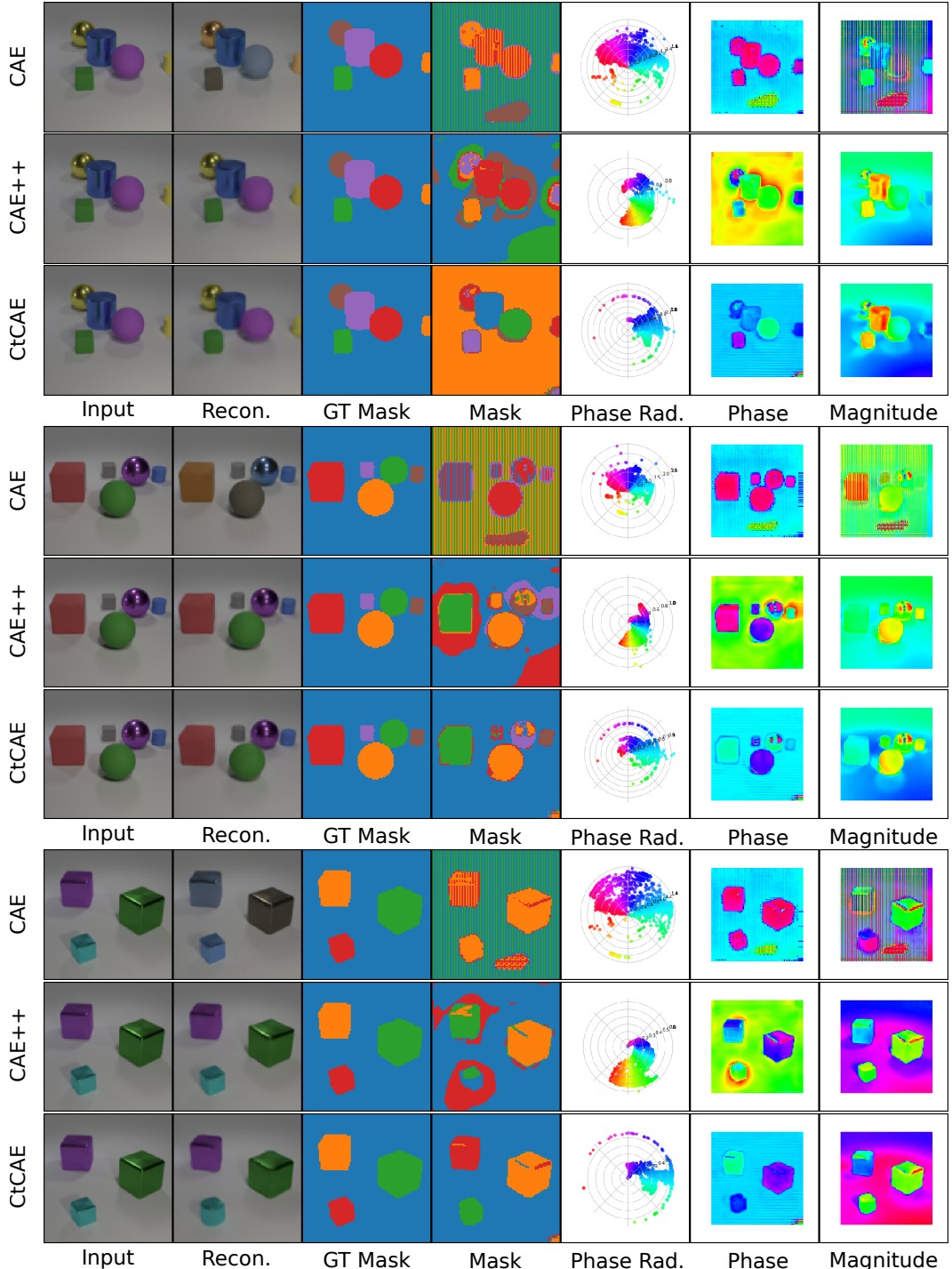

Figure 12: Visualization on grouping performance for CAE, CAE++ and CtCAE on CLEVR dataset. We are particularly interested in the phase space spread (column 5) here. CAE has a very large spread in phase values (column 5), but almost random object discovery performance, often grouping all objects together. But if we see the actual phase values assigned to objects (column 6), we observe that CAE tends to assign very similar phase values (e.g. all object phases' are in shades of magenta seen in column 6, rows 1, 4, and 7) to regions belonging to different objects whereas the corresponding maps show better separation in CAE++ and much superior in CtCAE. This explains better grouping performance (column 4) shown by CAE++ and CtCAE over the vanilla CAE.

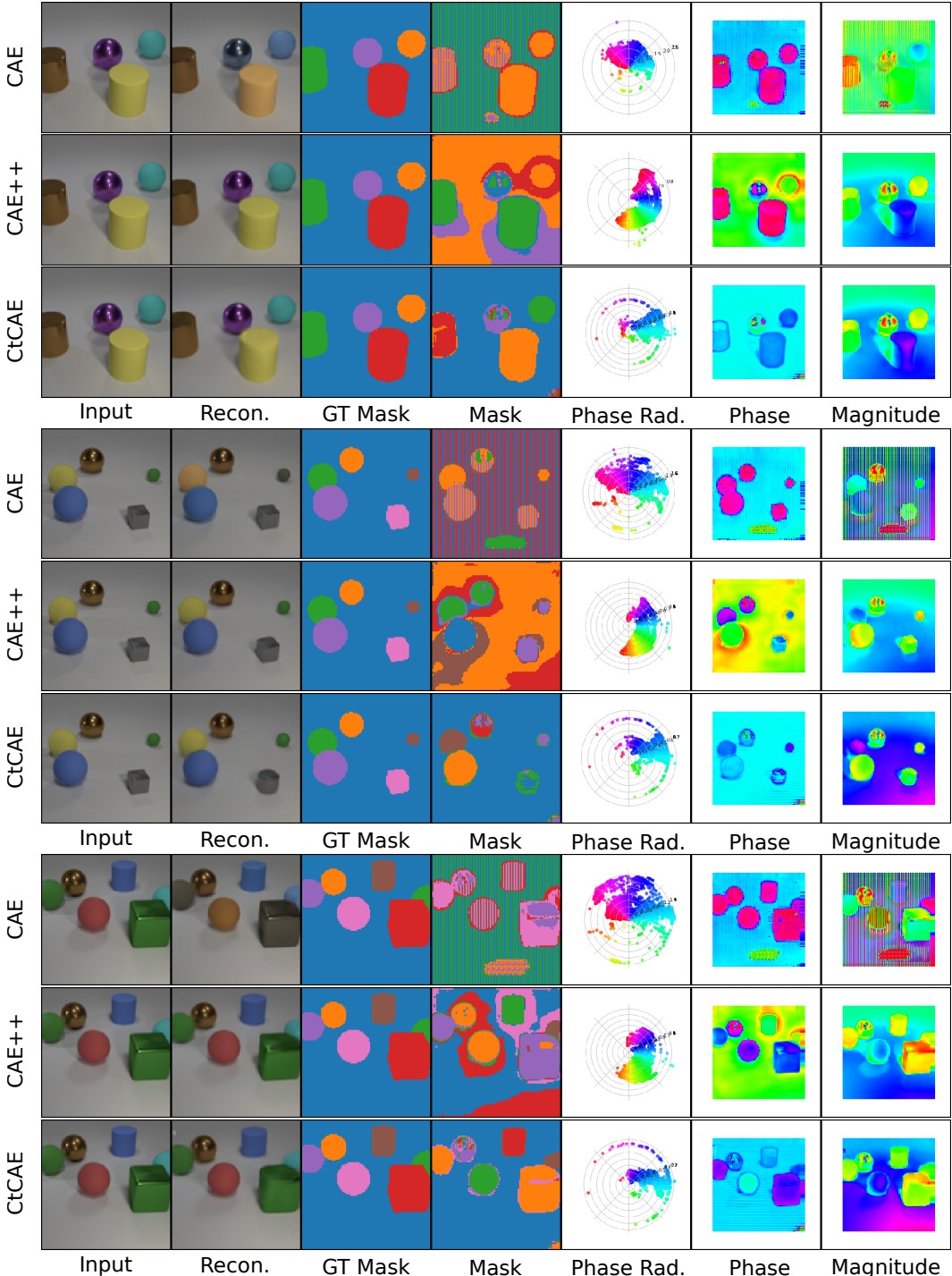

Figure 13: Visualization on grouping performance for CAE, CAE++ and CtCAE on samples with 4, 5 and 6 objects from the CLEVR dataset. CAE consistently shows poor object discovery, often grouping all objects together. CAE++ improves grouping substantially over the CAE baseline, but also struggles in certain scenarios. For example, in the scene with 4 objects it groups two different objects together, whereas for scenes with 5 and 6 objects on two samples here we see that it wrongly groups two objects together. CtCAE improves on these grouping failures of CAE++. CtCAE correctly discovers all objects in the scene with 4 and 5 objects, even though it makes an error grouping two objects together in the hardest example scene containing 6 objects.

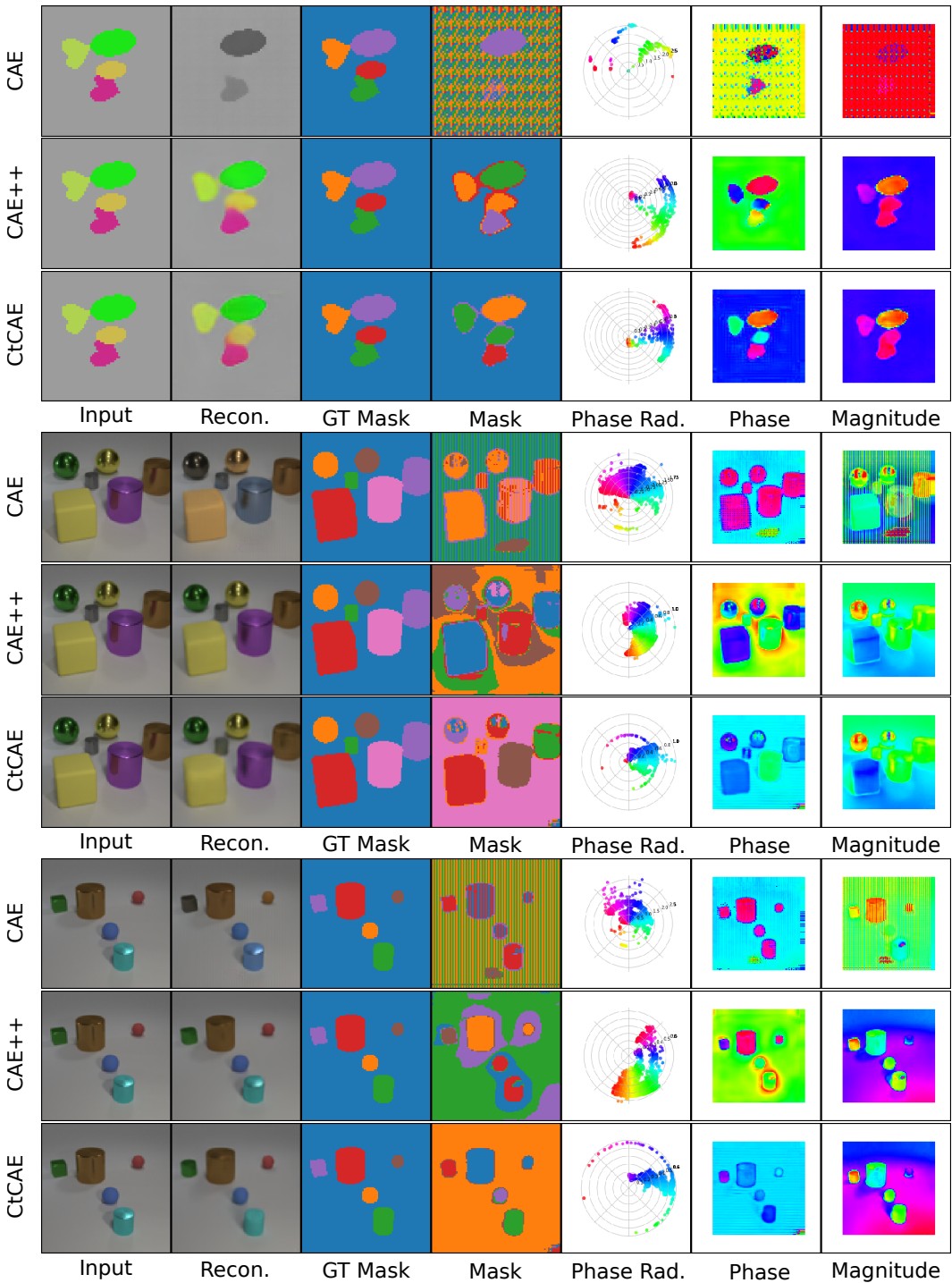

Figure 14: One of the limitations (failure modes) we observe is that in some images containing many objects, with some of them having the same (or similar) colors, our CtCAE model groups two such objects into the same cluster. Even in this example, CtCAE still clearly outperforms CAE++, which in turn outperforms the CAE baseline.

