# OpenReview forum: "Contrastive Training of Complex-Valued Autoencoders for Object Discovery"
_NeurIPS.cc/2023/Conference — NeurIPS 2023 poster_

### Official Review · Reviewer_pFdh · 2023-06-22

**Soundness:** 4 excellent
**Presentation:** 4 excellent
**Contribution:** 4 excellent
**Rating:** 8
**Confidence:** 4

**Summary:**

This paper improves the state-of-the-art in synchrony based object centric learning methods. The current SoTA is the Complex-valued AutoEncoder (CAE). But CAE is unable to handle even the Tetrominoes dataset. The paper presents CAE++ and CtCAE.

CAE++ = CAE + minor architectural improvements

CtCAE = CAE++ + Contrastive loss term.

Experiments show that CAE++ and CtCAE are the first synchrony based methods that can handle colour images, and CtCAE can handle up to 6 objects per image.

The contrastive loss term in CtCAE is intended to make the object clusters well separated in phase space. This is then evaluated in Table 2 by measuring inter-cluster and intra-cluster distances. The rest of the paper measures ARI and FG-ARI on Tetrominoes, dSprites, and CLEVR datasets. Ablation experiments are used to justify the design choices.

**Strengths:**

Although this line of work is niche, it is promising and engages scientific curiosity among many readers. The paper advances this line of work closer to the overall SoTA among object centric learning methods; although there is still a large gap here.

The method is easy to understand and the paper is well written.

The experiments cover most design choices and compare against CAE, the obvious baseline. The experiments also address important questions such as the robustness across varying number of objects and generalization to more objects. Error bars are included in all tables.

**Weaknesses:**

Please incorporate the following minor improvements
- In line 175, it is claimed that performance is closely matched at 32x32. This is not true for dSprites ARI-FULL is 0.90 vs 0.68. I believe it may be sufficient to say that the methods are ordered the same way across resolutions.
- In Table 2, please indicate in row 1 whether higher is better (up arrow) or down is better (lower arrow) to aid interpretation. In some cases numbers have been bolded despite overlapping error bars. Are these statistically significant?
- In lines 299-305, consider citing Odin (https://arxiv.org/abs/2203.08777) which does pixel level BYOL followed by kmeans to get objects. It is kind of a contrastive object centric method (although mean-teacher distillation is probably the right word for BYOL).
- Some background explanation about the encoder and decoder and where the kmeans happens to get masks would make it even easier to understand the method, for readers not familiar with CAE.

The main drawback is that slot based methods (DINOSAUR, https://openreview.net/forum?id=b9tUk-f_aG) are now working on real world data whereas Synchrony based methods are yet to make the break into textured images. But I think this paper is still relevant and that future developments in this line of work may catch-up to Slot based methods.

**Questions:**

I've asked my questions in the weaknesses section.

Additionally a sweep over the \beta parameter would be nice.

**Limitations:**

The authors have adequately addressed the weakness although the empirical gap between slot based and synchrony based methods should be made clearer in my opinion.

I do not see any potential negative societal impact of this work.

---

> ### Author Rebuttal · Authors · 2023-08-09
>
> We thank the reviewer for providing valuable feedback on our work and for the highly positive comments on the strengths of our contributions. We are very happy to hear that the reviewer found the method easy to understand, the paper to be well written and the experiments thorough.
>
> Below are our responses to your questions/comments:
>
> *“ In line 175, it is claimed that performance is closely matched at 32x32. This is not true for dSprites ARI-FULL is 0.90 vs 0.68. I believe it may be sufficient to say that the methods are ordered the same way across resolutions.”*
>
> We thank the reviewer for highlighting this error. We will make the requested change by indicating the broader point that the various models continued to be ordered in the saw way across resolutions.
>
> *“In Table 2, please indicate in row 1 whether higher is better (up arrow) or down is better (lower arrow) to aid interpretation.”*
>
> Thank you for this suggestion, we will make the requested change to Table 2 in the updated manuscript.
>
> *“In lines 299-305, consider citing Odin (https://arxiv.org/abs/2203.08777) which does pixel level BYOL followed by kmeans to get objects."*
>
> We thank the reviewer for pointing out this relevant work. We will cite and discuss this work in the related work section under “Contrastive Learning with object-centric models”.
>
> *“ The main drawback is that slot based methods (DINOSAUR, https://openreview.net/forum?id=b9tUk-f_aG) are now working on real world data whereas Synchrony based methods are yet to make the break into textured images.”*
>
> We share the view of the reviewer (noted in lines: 312-313) that the current synchrony-based family of models lag behind state-of-the-art slot-based approaches to real-world visual complexities. They have received far less attention by the research community who have directed significant algorithmic/engineering efforts over the last several years to scale them from simple multi-object datasets (MNIST/Shapes/Tetrominoes) to the current visual complexities (MoVi-C/E). We hope the research community invests a similar amount of effort towards scaling synchrony-based models as well.
>
> Regarding where the ‘masks’ used by K-means for clustering are obtained, we take the phases at the decoder output layer as inputs for the clustering algorithm to compute the object assignments (please refer to L101-103).
>
> *“Additionally a sweep over the $\beta$ parameter would be nice.”*
>
> We initially evaluated several values for the $\beta$ parameter such as [0.01, 0.001, 0.0001, 0.00001, 0.000001] and quickly settled to the value of 0.0001 (kept constant across all experiments in this work) as it led to better training and grouping performance across different datasets. As per your request, we additionally ran a sweep for $\beta \in [0.01, 0.001, 0.0001, 0.00001, 0.000001]$ on the 32x32 dSprites dataset.
>
> Here are the ARI scores we obtained for various values of $\beta$ (shown on the first column below):
>
> 0.01         - ARI-FG: 0.20$\pm$0.06; ARI-FULL: 0.14$\pm$0.04
>
> 0.001       - ARI-FG: 0.43$\pm$0.05; ARI-FULL: 0.66$\pm$0.11
>
> 0.0001     - ARI-FG: 0.48$\pm$0.03; ARI-FULL: 0.68$\pm$0.13
>
> 0.00001   - ARI-FG: 0.48$\pm$0.07; ARI-FULL: 0.53$\pm$0.05
>
> 0.000001 - ARI-FG: 0.46$\pm$0.04; ARI-FULL: 0.49$\pm$0.11
>
> This extra sweep confirms that $\beta=0.0001$ (used throughout the work) is a good choice.
>
> Please let us know if there are any concerns you have raised that our responses have not yet addressed satisfactorily.

---

> > ### Comment · Reviewer_pFdh · 2023-08-13
> >
> > Thank you for the sweep over \beta and addressing the remaining concerns. I continue to support the acceptance of this paper.

---

### Official Review · Reviewer_bBXA · 2023-07-02

**Soundness:** 2 fair
**Presentation:** 4 excellent
**Contribution:** 3 good
**Rating:** 5
**Confidence:** 4

**Summary:**

CAEs hold potential to address several issues associated with slot-based representation. They may provide flexible object granularity, enable the extraction of part-whole hierarchies as needed, and provide faster training speeds. However, testing of the original CAE was confined to grayscale images and a small number of objects (2-3). This paper explores the question: how can we scale up CAEs beyond this initial scope?

For this, the paper takes the following approach:
1. Discovers certain architectural modifications that make original CAE perform much better on more complex datasets, in this case, Tetris, CLEVR, and dSprites.
2. However, noting that this is not enough, the paper proposes a novel contrastive loss, which improves the segmentation performance even further when added to the original reconstruction loss of CAE.

**Strengths:**

1. Useful ablations show that architectural optimizations can be important to achieve the full potential of CAE. A good thing about some of these optimizations is that they actually relax and don’t complicate the original architecture e.g., by removing sigmoid activation.
2. Empirically show higher storage capacity in terms of the number of objects than CAE.
3. Paper is largely well-written.

**Weaknesses:**

1. While it is interesting that contrastive loss helps the segmentation performance, it somehow does not provide me new insights about the CAE framework itself. For instance, it doesn’t illuminate why separability was poor in the original CAE in the first place. As such, I am a bit worried that the proposed solution addresses a symptom rather than the root cause.
2. Line 215: *“The results in Table 2 indicate that indeed, intra-cluster distance (the 216 last column) is smaller for CtCAE than CAE++ for every dataset, especially for the most difficult ones”* It is actually difficult to conclude this from Table 2 considering the large standard deviations of these measurements. My suggestion would be to tone down this claim and/or confine this result to the appendix. Regarding the separability results more broadly, it is not clear whether separability in the representation space (as defined in this paper) is actually important with respect to the things we may eventually care about e.g., downstream performance or other tasks.
3. (Optional) It would be good to also report performances of some slot-based baselines, not for strict comparison, but to help the reader position the current performance in a more complete perspective. I see a line mentioned about it in the conclusion but without a pointer to a supporting result.

**Questions:**

1. Why is CAE performance not reported in Table 3?
2. Why not apply the contrastive loss on all layers rather than just the output of the encoder and the decoder? It appears a bit arbitrary to choose two specific layers to apply this loss.

**Limitations:**

Yes, the paper discusses the limitations relative to current slot-based methods, but one should also consider the potential benefits of the CAE family of models.

---

> ### Author Rebuttal · Authors · 2023-08-09
>
> We thank the reviewer for giving us valuable feedback and critiques on our manuscript. We’re happy to hear that the reviewer found our paper to be well-written and ablation studies on architectural design to be useful and simplifying. Below are our responses to your questions/comments:
>
> *“... does not provide me new insights about the CAE framework itself. For instance, it doesn’t illuminate why separability was poor in the original CAE in the first place.”*
>
> Our intuition for designing the contrastive loss stems from training the CAE model with solely the pixel-level reconstruction objective provided insufficient optimization pressure to guide the learning of bias terms ($b_{\psi}$) to discover the various independent factors (objects) in the scene. Beyond that, we do not claim to provide further insights into why just the phase interference mechanisms in CAE activations are insufficient to allow it to learn meaningful groups on more complex datasets (like the ones tested here). We believe this could be an interesting analysis for future work.
>
> *“... difficult to conclude this from Table 2 considering the large standard deviations of these measurements.”*
>
> We thank the reviewer for highlighting this aspect of the results in Table 2. We agree with the reviewer, and will reduce the strong claim in the updated manuscript as follows: The results in Table 2 show that, the mean intra-cluster distance (the last column) is smaller for CtCAE than CAE++ on two (dSprites and CLEVR) of the three datasets. However, we believe it is important to retain Table 2 in the main text due to its relevance to the proposed contrastive loss.
>
> *“... it is not clear whether separability in the representation space (as defined in this paper) is actually important with respect to the things we may eventually care about e.g., downstream performance or other tasks.”*
>
> Separability improves the ability of the model to group/store independently the representations of various objects in a scene. This separation of information internally allows the model to generalize better in compositional scenarios such as varying the number of objects, novel combinations of known objects etc by avoiding the “superposition catastrophe” problem [1]. The ability to separately maintain the information of different object instances also allows us to compute relational factors between them useful to solve tasks that involve object-level reasoning. This is the same motivation behind the task of perceptual grouping with slot-based models [1]. We do not make any specific claims about how “separability” is positively correlated to downstream task performance in our work.
>
> *“(Optional) It would be good to also report performances of some slot-based baselines, not for strict comparison, but to help the reader position the current performance in a more complete perspective. I see a line mentioned about it in the conclusion but without a pointer to a supporting result.”*
>
> We thank the reviewer for this suggestion (as was also noted by Reviewer BSdH). We agree to this recommendation and will update the manuscript adding the grouping results for a baseline SlotAttention model to Table 1.  For reference, SlotAttention achieves ARI-FG of $98.8\pm0.3$ for CLEVR, $91.3\pm0.3$ for dSprites and $99.5\pm0.2$ for Tetrominoes (results from Emami et al. [2]).
>
> *“Why is CAE performance not reported in Table 3?”*
>
> This was because the CAE baseline does not learn any meaningful grouping at all on dSprites and CLEVR as can be seen in Table 1. It achieves very low ARI-FG and ARI-FULL scores (near zero in several cases). Therefore, we decided to remove it from the results in Table 3 as at this very low level of grouping performance we cannot draw any meaningful conclusions about its ability to generalize to different numbers of objects in the scene.
>
> *"Why not apply the contrastive loss on all layers rather than just the output of the encoder and the decoder? It appears a bit arbitrary to choose two specific layers to apply this loss."*
>
> This is a useful suggestion (as was noted by Reviewer QpC7 as well). We did experiment with various configurations applying the contrastive loss to different layers of both the encoder and decoder modules, including the reviewer’s recommendation of applying the contrastive loss to every convolutional layer. However, we observed that adding more such contrastive loss terms did not help improve the grouping performance and so we utilized the minimal (but still the best performing) choice of contrasting only the encoder and decoder outputs.
>
> Please let us know if there are any concerns that the reviewer raised that our responses have not addressed satisfactorily. If the reviewer found our response useful, could the reviewer please consider increasing the score. Thank you very much!
>
> References:
>
> [1] Greff et. al, “On the Binding Problem in Artificial Neural Networks”, 2020.
>
> [2] Emami et. al, “Efficient Iterative Amortized Inference for Learning Symmetric and Disentangled Multi-Object Representations”, ICML 2021.

---

> > ### Comment · Reviewer_bBXA · 2023-08-18
> > **Thank You**
> >
> > My biggest concern is still whether contrastive loss is the right direction to go from the original CAEs. To me, new contrastive loss somehow contradicts the main promise of CAEs. CAEs are purported to take us away from hard clustering decisions about what objects are --- to leave flexibility for the downstream user if they want to attend to a part or a whole by giving them a "map of phases".
> >
> > By enforcing separability by force (in this case by asking pixels/patches with similar visual information to become close in phase space), we are going back to square one --- back to hard clusters of slot attention.
> >
> > I believe in the promise of CAEs but I think a good solution to scaling CAEs may come from a better understanding of the original CAEs (which authors say will fall in the realm of future work) and also asking why hard separability should even be desired.
> >
> > I am fully onboard about going from CAE to CAE++ which introduces modifications to aid optimization of the model. But for the reasons mentioned above, I am somehow finding it difficult to convince myself to go from CAE++ to CtCAE.
> >
> > I will maintain my score for now but I may be a bit less inclined towards acceptance than before.

---

> > > ### Author Response · Authors · 2023-08-20
> > >
> > > Thank you very much for your comment and initiating this discussion on the high-level motivation behind our work.
> > >
> > > > *“To me, new contrastive loss somehow contradicts the main promise of CAEs.”*
> > > > *“By enforcing separability by force (in this case by asking pixels/patches with similar visual information to become close in phase space), we are going back to square one --- back to hard clusters of slot attention.”*
> > >
> > > Unlike slots, our contrastive loss does not impose activations to form a fixed set of clusters, thereby the core essence of synchrony-based approaches—flexible and graded quality of clusters—is still preserved. The entire object is not simply assigned a single phase value. It is still up to the downstream user to extract clusters at the granularity level needed by the specific task, on-demand, given the “map of phases” using the appropriate level of quantization of the continuous phase map. The “hard” decision about what the “right” objects are is still left to the specifics of the clustering procedure used which can be conditioned based on the downstream task and is not hard-coded during the training phase.
> > >
> > > > *“good solution to scaling CAEs may come from a better understanding of the original CAEs”*
> > >
> > > We agree with the reviewer, but “better understanding of the original CAEs” itself seems to be an important open-ended research challenge. This difficulty is also partially due to the lack of maturity of these models. The CAE architecture and earlier works by Reichert et. al [37] (following the reference numbering in our paper) motivated the binding mechanism in synchrony-based models simply via phase interference (addition of complex-valued activations). In practice, it still required the use of the gating mechanism which modulates the output response of a unit as function of the phase difference such that even activations with opposite phases do not completely cancel out their magnitude responses. This architectural change was necessary for these models to learn meaningful structure in the phase space. Simply training these models to reconstruct pixels without this gating mechanism was insufficient to drive emergence of any meaningful structure in their phase space (ablation without gating in Table 2 in Löwe et. al [33]). This highlights a “gap” between the conceptual motivations in synchrony-based models and the practical engineering design choices required to get this class of models to train/learn meaningful visual structure.
> > >
> > > At this stage, general explorations in the space of both architectures and training objectives seem necessary to gain empirical knowledge about these models. This is especially the case when there is no guarantee that the mainstream pixel-level reconstruction loss is enough for the model to behave as expected; in fact, the old literature on the synchrony models even used a “supervised learning approach” (see Mozer et al. [35]) to encourage phase learning/spreading. Further, such explorations could also go hand in hand with future, better understanding of the model architecture.
> > >
> > >
> > > > *“asking why hard separability should even be desired”*
> > >
> > > We agree with the reviewer that “hard” separability is not desirable apriori. But, we need some priors about the independence in the visual input that must be injected either via the architecture or the training objective for the system to discover these independent factors in its phase map. Therefore, here we have chosen to create minimal optimization pressure for the network to discover the “right” independent factors in the visual scene by our regularization objective.

---

### Official Review · Reviewer_BSdH · 2023-07-05

**Soundness:** 3 good
**Presentation:** 2 fair
**Contribution:** 3 good
**Rating:** 7
**Confidence:** 3

**Summary:**

This paper presents a collection of small modifications of the Complex-valued AutoEncoder architecture leading to improved reconstruction/segmentation performance on simple RGB datasets; a contrastive learning objective is introduced to further improve the performance.

**Strengths:**

- The paper's story is easy to follow, and the motivation is clearly stated.
- Architectural choices are justified through ablation studies.
- Experimental results are averaged over multiple random seeds.

**Weaknesses:**

- The captions of figures/tables could be improved to make it easier for readers to understand what is displayed. For example, Figure 1 is hard to understand without reading the relevant section.
- Table 1: While I see that this is a comparatively new line of research on new approaches for object-centric learning, I'd still appreciate it if the authors compared to state-of-the-art models so that readers get a feeling for how far the gap still is/how small it has become.

**Questions:**

- Eq. 2: A short note on why this separation is used and not one that uses the magnitude and phase could be helpful.
- L134: Moving the reference to Figure 1 to the previous paragraph might make it easier to follow the general idea of this loss.
- Eq. 6: Is this the loss for a single input image, meaning that the loss is then averaged over a batch? Or are the negative/positive samples chosen across a batch? If it's not across the entire batch, what speaks against doing this?
- Figure 2: An explanation of what the last three columns show/what one can learn from looking at them can be useful for readers.
- L218: Please elaborate on why the minimum distance is more relevant than the average.

**Limitations:**

While the authors mention the strongest limitation, a more extensive discussion of this approach's limitations is missing.

---

> ### Author Rebuttal · Authors · 2023-08-09
>
> We thank the reviewer for the valuable feedback and suggestions to improve our work. We are happy that the reviewer found the manuscript easy to follow and the main ideas presented to be clearly motivated. Below are our responses to the comments/questions from the reviewer:
>
> *“ … captions of figures/tables could be improved to make it easier for readers to understand what is displayed.”*
>
> Indeed, due to the space limitation, we had to avoid redundancy between the caption and section text. To improve this, we will add a link in the caption that refers to the corresponding text in the section (which provides the necessary contextual information), wherever appropriate. We thank the reviewer for pointing this out.
>
> *“ I'd still appreciate it if the authors compared to state-of-the-art models so that readers get a feeling for how far the gap still is/how small it has become.”*
>
> We thank the reviewer for this suggestion (as was also requested by reviewer bBXA). We agree to this recommendation and will update the manuscript adding the grouping results for a baseline SlotAttention model to Table 1. For reference, SlotAttention achieves ARI-FG of $98.8\pm0.3$ for CLEVR, $91.3\pm0.3$ for dSprites and $99.5\pm0.2$ for Tetrominoes (results from Emami et al. [3]).
>
> *“ Eq. 2: A short note on why this separation is used and not one that uses the magnitude and phase could be helpful.”*
>
> This use of the cartesian form (real and imaginary components of a complex-valued activation) in a layer’s forward pass was the formulation introduced by Reichert and Serre [1] which has been maintained in Löwe et. al [2] as well.
>
> *“ L134: Moving the reference to Figure 1 to the previous paragraph might make it easier to follow the general idea of this loss.”*
>
> We thank the reviewer for this suggestion and agree to move the reference to Figure 1 to the paragraph above to improve the readability.
>
> *“ Eq. 6: Is this the loss for a single input image, meaning that the loss is then averaged over a batch? Or are the negative/positive samples chosen across a batch? If it's not across the entire batch, what speaks against doing this?”*
>
> The contrastive loss is computed on a per-image basis and averaged across the samples in a batch. The positive and negative pairs are extracted from within one image. This is because we would like to enforce the phases of pixels within a single image to be far apart or close depending on the objects they belong to. The phase values of objects across images are not relevant to achieve this instance-level grouping of objects within a scene. For example, given 2 images: img_1: contains a red ball and green cube || img_2: red ball, green cube and yellow triangle. Ideally we would want phases of the ball and cube in img_1 to be separated by 180 degrees, whereas in img_2 their difference is ideally 120 degrees. The red ball and green cube cannot have the same absolute phase values across both these images. Rather, what matters more is the relative phase between objects within a single image (scene). Therefore, it would be incorrect to contrast object instances across image samples in a batch to achieve instance-level grouping.
>
> *“ Figure 2: An explanation of what the last three columns show/what one can learn from looking at them can be useful for readers.”*
>
> We thank the reviewer for this suggestion. We will include descriptive text for e.g., “column 5 shows the radial plot with the phase values from $-\pi$ to $\pi$ radians (colors correspond to phase angles), column 6 shows the raw phase values (in radians) averaged over the 3 output channels as a heatmap (where colors correspond to the colors from Phase Rad. column 5) and column 7 shows magnitude component of the complex-valued outputs” in the updated manuscript to assist the reader’s interpretation of the last three column's contents in Figure 2.
>
> *“ L218: Please elaborate on why the minimum distance is more relevant than the average.”*
>
> Separability indicates how difficult it is for the phase pattern of one object to interfere with another. During the evaluation procedure to extract discrete object assignments from the continuous phase maps, a clustering procedure is applied on the phase values. Our clustering procedure (KMeans) uses the centroid that is closest (minimum) distance to assign a phase value as being part of that cluster. Therefore, minimum distance is more relevant to the accuracy of the extracted object assignments through clustering than average distance.
>
> *“ … a more extensive discussion of this approach's limitations is missing.”*
>
> We would like to highlight the qualitative failure modes of CAE++/CtCAE discussed in lines: 259-263. We visualize and qualitatively examine these failure modes in Figure 4 in the main text as well as provide more such qualitative samples shown in Figure 14 in Appendix C.  Please let us know if you have some specific recommendations in-mind for the discussion of limitations that we have not included in the paper.
>
> Please let us know if there are any concerns remaining that the reviewer raised that our responses have not sufficiently addressed.
>
> References:
>
> [1] Reichert et. al, “Neuronal Synchrony in Complex-Valued Deep Networks”, ICLR 2014.
>
> [2] Löwe et. al, “Complex-Valued Autoencoders for Object Discovery”, TMLR 2022.
>
> [3] Emami et. al, “Efficient Iterative Amortized Inference for Learning Symmetric and Disentangled Multi-Object Representations”, ICML 2021.

---

> > ### Comment · Reviewer_BSdH · 2023-08-13
> >
> > Thank you for responding to my question and addressing the points I raised. As mentioned in my original review, I believe that while the ansatz presented in this work is currently performing worse than other established (and long-refined approaches), there is nevertheless a value in finding new approaches and would, hence, recommend accepting this paper.

---

### Official Review · Reviewer_QpC7 · 2023-07-05

**Soundness:** 1 poor
**Presentation:** 3 good
**Contribution:** 3 good
**Rating:** 6
**Confidence:** 4

**Summary:**

This paper proposes several improvements over the Complex AutoEncoder (CAE) to scale it to more complex multi-object datasets. The CAE is a synchrony-based model that learns object-centric representations by injecting complex-valued activations into every layer of its architecture, but it is only applicable to grayscale images with up to 3 objects. In this paper, two modifications to the original architecture and a novel auxiliary loss function are proposed that make the CAE applicable to RGB datasets with more than 3 objects.

**Strengths:**

Overall, the paper is very well written and easy to follow. The introduction provides a good overview over the relevant research and a convincing motivation for the proposed approach. The methods section describes the proposed improvements and novel contrastive loss clearly. The results section is well-structured, and the related works section embeds the proposed research well into the existing research landscape.

The proposed contrastive loss seems like an elegant solution towards improving the "separability" of objects in the phase space. The complex-valued features contain two separate types of information (feature presence in the magnitudes and feature binding in the phases), and the proposed contrastive loss smartly utilizes this separation.

The provided ablation studies provide interesting insights into the proposed model improvements and design choices.

**Weaknesses:**

1) The main problem with the paper is that it seems to evaluate the models incorrectly. As described in section C.1 - "RGB Images" of the paper proposing the CAE [1], it cannot simply be applied to RGB images due to its evaluation method. During evaluation of the CAE, as well as in the proposed approach, features with small magnitudes are masked out, as they may exhibit increasingly random orientation values. However, this rescaling may lead to a trivial separation of objects in RGB images. For an image containing a red and a blue object, for example, the reconstruction will assign small magnitudes to the color channels that are inactive for the respective objects. By masking out values with small magnitudes, objects will subsequently be separated based on their underlying color values, rather than their separation as learned by the model. As far as I can tell, the paper does not describe any mechanism that would prevent the evaluation from doing this. I believe this effect is even visible in the qualitative examples. In Figure 3, the phase image of the CtCAE model on the dSprites example does not seem to clearly separate the purple and orange objects. The mask, however, separates them perfectly. In my eyes, this is only possible if the evaluation procedure makes use of additional (color) information besides the pure phase output. Overall, this makes it unclear whether the presented results are valid.

2) Eq. 6: When measuring the cosine distance between the phase feature vectors, extra care would be necessary to take into account their circular nature. Otherwise, two phase vectors $a=(0,0,0)$ and $b=(2 \pi, 2 \pi, 2 \pi)$ would have a large cosine distance to one another even though they represent the same angles. Has this been taken into account?

3) It would be interesting to see an evaluation of how the proposed improvements influence the separation of objects in the latent space of the model. Ultimately, the goal of object discovery is to create object-centric representations within the model, but the paper currently only evaluates the separation at the output.

Minor points:
- line 25: not all slot-based method work with iterative attention procedures
- Personally, I would move the CAE overview into a separate Backgrounds section
- line 90: $m_\psi$ can technically not be called a magnitude, as it may contain negative values.
- Eq. (6): Is there no exp() applied to the cosine distances, such that the equation would correspond to a Softmax?
- Using a dimensionality of 32x32 for Tetrominoes changes the shapes of the objects. While at the original dimension, all objects are made up from perfect squares, these squares get squeezed randomly when resizing the images. Thus, it would be better to make use of the original input dimension.



---

[1] Löwe, S., Lippe, P., Rudolph, M., & Welling, M. (2022). Complex-valued autoencoders for object discovery. arXiv preprint arXiv:2204.02075.

**Questions:**

1) As described above, I believe the main weakness of this paper is its evaluation procedure. If the authors could come up with a solution to the described problem or convince me that this problem does not apply to their setup, I'd be happy to increase my score.


Other questions:

2) Why do you not apply the contrastive loss to every (convolutional) layer of the architecture?

3) Do you have an explanation as to why the standard deviation is so large for some of the results of the CtCAE (Table 1)?

**Limitations:**

Limitations are adequately addressed.

---

> ### Author Rebuttal · Authors · 2023-08-09
>
> We thank the reviewer for providing valuable feedback on our work. We’re glad that the reviewer found our paper well-written, the proposed contrastive loss well-motivated and an elegant solution to the problem and the experiments/ablation studies insightful.
>
> Below are our responses:
>
> *”main problem with the paper is that it seems to evaluate the models incorrectly.”*
>
> It is true that in our setting, certain color information may contribute to the separation process, and the example by the reviewer provides an illustration. However, the reviewer's claim "By masking out values with small magnitudes, objects will subsequently be separated based on their underlying color values, rather than their separation as learned by the model." cannot be true, since CtCAE largely outperforms CAE++ (see Table 1) while both of them have a similar (almost perfect) reconstruction loss and (of course) are evaluated in the same way. This clearly indicates that the potential issue pointed out by the reviewer only has a marginal effect in practice in our setting, and that separation learned by the model is crucial to achieve good performance. In fact, “pure” RGB colors (as in the reviewer's example “containing a red and a blue object”) very rarely occur in images from the multi-object datasets (dSprites, CLEVR) used in this work. The percentages of pixels with a magnitude below 0.1 are 0.44%, 0.88%, and 9.25% for CLEVR, dSprites, and Tetrominoes respectively. While 9% on Tetrominoes is not marginal, we empirically observe that this is not an issue in practice, since we observe many examples in Tetrominoes where two or three blocks with the same color get grouped into different clusters by either CAE++ or CtCAE (see three yellow blocks in Figure 3, two magenta blocks in Figure 8, two magenta blocks in Figure 9), demonstrating that the separation is clearly not simply based on color values of pixels. Therefore, while we agree that our object extraction process should ideally be fixed for these edge cases, we leave that as an investigation for future work.
>
> *”In Figure 3, the phase image of the CtCAE model on the dSprites example does not seem to clearly separate the purple and orange objects ....... only possible if the evaluation procedure makes use of additional (color) information besides the pure phase output.”*
>
> We would like to emphasize that the ‘Phase Img.’ (shown in columns 3/6/9 of Figure 3) is the average of the 3 output phase values plotted as a heatmap. Due to this averaging it cannot distinguish between different phase values eg. [0.1*$\pi$, 0.5*$\pi$, 0.2*$\pi$], [0.2*$\pi$, 0.1*$\pi$, 0.5*$\pi$], [0.5*$\pi$, 0.2*$\pi$, 0.1*$\pi$]. Therefore the visual discrepancies between a ‘Mask’ and corresponding ‘Phase Img.’ are due to the use of the “average” phase plot. (Note that for the evaluation the KMeans clustering is applied on all 3 phase values (of a pixel) at the decoder output and is therefore able to discriminate between these different phase values leading to a correct prediction of the ‘Mask’.) We understand this aspect of the visualization is not explained and will add a note in the updated manuscript. Thank you for drawing our attention to this.
>
> *”Eq. 6: When measuring the cosine distance between the phase feature vectors, extra care would be necessary to take into account their circular nature.”*
>
> We investigated a variant which accounted for the circularity of phase space but it did not improve the results. We observed in practice that phases are initialized from zero and rarely go beyond the range [-0.75*$\pi$, 0.75*$\pi$] (as observed in Figure 2 and 3).
>
> *”.. would be interesting to see an evaluation of how the proposed improvements influence the separation of objects in the latent space of the model ... currently only evaluates the separation at the output.”*
>
> Evaluating the separation of objects only at the output level follows the method used by the original CAE model. It allows a fair comparison of grouping performance learned by the baseline models and our proposed variants. We agree with the reviewer that this idea of evaluating object separation at several layers within the synchrony-based model is interesting and valuable but it is outside the scope of our current work.
>
> *”Why do you not apply the contrastive loss to every (convolutional) layer?”*
>
> We experimented with various configurations applying the contrastive loss to different layers of the encoder and decoder, including applying the loss to every layer. We observed that adding more such loss terms did not help improve grouping performance and so we utilized the minimal (but still the best performing) choice of contrasting only the encoder and decoder outputs.
>
> *Do you have an explanation as to why the standard deviation is so large for some of the results of the CtCAE (Table 1)?*
>
> Regarding the std-dev for CtCAE in Table 1, please note that the comment about the larger std-dev for CtCAE compared to other models is only true for the ARI-FULL score on CLEVR dataset. Please note that on some dataset variants like dSprites (32x32) or CLEVR (96x96) the std-dev for CtCAE is lower and in other cases it is comparable to the std-dev of CAE++.
>
> *”On minor points.”*
>
> We thank the reviewer for the suggestion to move the CAE description to a background section and agree to make this change. Thank you for pointing out that $m_{psi}$ cannot technically be called a magnitude due to potential negative values, we will correct this. We will correct the statement (line 25) to include a broader reference (not just iterative attention) to segregation mechanisms in slot-based models. Finally, thank you for spotting the error in Eq. 6. Indeed, we exponentiate the cosine distances resulting in a Softmax term. We apologize for this typo and will correct it in the updated manuscript.
>
> We hope our responses resolve the major comments/questions posed by the reviewer. Please let us know if any other questions are yet to be addressed.

---

> > ### Comment · Reviewer_QpC7 · 2023-08-10
> >
> > Thank you for your reply. Besides the two concerns described below, all my questions have been adequately addressed.
> >
> > 1) ”main problem with the paper is that it seems to evaluate the models incorrectly.”
> >
> > While it is true that the current form of evaluation has a neglible influence on the comparison between the CAE++ and CtCAE models, the same cannot be said when comparing against other models. For one, the CAE achieves a much poorer reconstruction error, which will limit its ability to benefit from any color information that may contribute to the evaluation. Most importantly, the current setup renders a comparison unfair with evaluation procedures in which color information will never influence the segmentation result. For example, in response to other reviewers, the authors stated that they will add a SlotAttention baseline to Table 1. SlotAttention's evaluation procedure is entirely independent of the color information in the images, however, making a fair comparison to the current results impossible. Similarly, this may influence future work when trying to compare against the results presented in this paper, or when trying to apply this model to other datasets with different pixel statistics. Thus, I would propose to include a suitable disclaimer in the paper that describes the limitations of the presented results and evaluation procedure.
> >
> > 2) "Evaluating the separation of objects only at the output level follows the method used by the original CAE model."
> >
> > The original CAE paper includes a (limited) evaluation of the feature separation in the latent space in Figure 5. Both in this evaluation, as well as in new work on synchrony-based models [1], the separation in the latent space seems to be limited to two objects. Considering that the proposed method claims to improve separatibility, it would be very interesting to see whether this applies to the latent space as well, and whether this would help to overcome the aforementioned limitation.
> >
> > [1] Löwe, Sindy, et al. "Rotating Features for Object Discovery." arXiv (2023).

---

> > > ### Author Response · Authors · 2023-08-12
> > >
> > > Thank you very much for your prompt response and your engagement!
> > >
> > >
> > > > *“While it is true that the current form of evaluation has a negligible influence on the comparison between the CAE++ and CtCAE models, the same cannot be said when comparing against other models. For one, the CAE achieves a much poorer reconstruction error, which will limit its ability to benefit from any color information that may contribute to the evaluation.”*
> > >
> > > Thank you for agreeing that this evaluation has negligible influence on the comparison between CAE++ and CtCAE models. We respectfully disagree that this puts CAE at a disadvantage. Firstly, if the method is unable to reconstruct images completely (CAE in Tetrominoes), it is intuitively expected it would have a low (zero) clustering score. Secondly, it can be seen that CAE reconstruction on CLEVR is close to CAE++ and CtCAE (e.g. Table 1 CLEVR 32x32 - CAE++ actually has the lowest MSE whereas CtCAE and CAE are close). Even in this case CAE has 5x lower ARI-FG score and 7x lower ARI-FULL score compared to CtCAE.
> > >
> > > > *”Most importantly, the current setup renders a comparison unfair with evaluation procedures in which color information will never influence the segmentation result. For example, in response to other reviewers, the authors stated that they will add a SlotAttention baseline to Table 1. SlotAttention's evaluation procedure is entirely independent of the color information in the images, however, making a fair comparison to the current results impossible.”*
> > >
> > > We would like to respectfully disagree on this point too. The main evaluation procedure for both classes of methods (slot-based and synchrony-based) are the same in the current literature: it is about measuring the ARI score given some extracted alpha mask. There are fundamental differences though in the way the masks are “extracted” from these two classes of models. In slot-based ones typically one takes the mixing alpha-mask at the decoder output. One could argue that in such methods taking the alpha mask of the latent attention would be a better option. Still it’s a design choice. On the other hand, synchrony-based methods don’t have such explicit masks, so one has to devise a way of generating them, and all methods so far in the literature use KMeans clustering to do so. In sum, different methods have different ways of extracting masks as part of their design choices, and this is not an issue. What matters is the final evaluation metrics.
> > >
> > > Having said this, we should keep in mind that the evaluation through ARI scores is not the end goal, but more a proxy when it comes to object-centric representation learning. The actual thing we care about is some downstream task that evaluates the usefulness of these representations, for example set prediction (as in the SlotAttention paper), question-answering task or using it inside an agent. Downstream task performance of the synchrony-based models is another avenue for future research in terms of modes of evaluation of the quality of synchrony-based object representations.
> > >
> > >
> > > > *”Similarly, this may influence future work when trying to compare against the results presented in this paper, or when trying to apply this model to other datasets with different pixel statistics. Thus, I would propose to include a suitable disclaimer in the paper that describes the limitations of the presented results and evaluation procedure.”*
> > >
> > > We fully agree that the optimal way of generating clustering masks for synchrony-based methods is an open research problem and we will add a disclaimer in the updated paper about the current limitations as requested by the reviewer.

---

> > > > ### Author Response · Authors · 2023-08-12
> > > >
> > > >
> > > > > *”The original CAE paper includes a (limited) evaluation of the feature separation in the latent space in Figure 5. Both in this evaluation, as well as in new work on synchrony-based models [1], the separation in the latent space seems to be limited to two objects. Considering that the proposed method claims to improve separatibility, it would be very interesting to see whether this applies to the latent space as well, and whether this would help to overcome the aforementioned limitation.”*
> > > >
> > > > We are aware of this evaluation in the CAE paper, however we find it limited for a few reasons. Firstly, after the CAE is trained, only the encoder is retained and the decoder is trained from scratch to reconstruct *individual* objects. We believe such an evaluation protocol is rather ad-hoc that the decoder network can learn to "game" and might not give us an exact insight into the perceptual information (in pixel space) stored in one phase cluster versus another. Furthermore, this evaluation requires clustering latent representations and then masking out features that are not part of a certain cluster. Due to imperfections of the clustering process this would potentially lead to flawed conclusions. We believe a more interesting evaluation of synchrony-based methods would be a set prediction task or a generalization task, for example learning to sum up numbers presented in the image, in particular on a dataset where there are more objects than in images on which the model was trained on. Therefore, although we do agree that the evaluation of the latent representations of CtCAE model in terms of its informational content would be interesting, we believe that (due to the above stated reasons) it would require careful thought into what the best way to do so and therefore requires a disproportionate amount of effort. For this reason, we leave it for future work.
> > > >
> > > > As a side note we believe RF [1] is not directly comparable to CtCAE since it either relies on depth masks to learn the grouping or uses powerful contrastively-trained models to extract the latent features to then learn largely *semantic* segmentation masks (multiple instances of “cows” or “trains” in Figure 2 or “bicycles” in Figure 16 or “bread slices” in Figure 17 from [1]). Moreover, RF [1] was posted to ArXiv on 1st June 2023, 14 days after the NeurIPS main paper submission deadline (May 17th) and 7 days after the supplementary deadline (May 26th). We feel it is quite unreasonable to expect authors to compare to such recent results/observations. We will cite RF [1] in the updated version and discuss these differences between the two works.

---

> ### Comment · Area_Chair_XvoA · 2023-08-17
> **AC request for clarification**
>
> Dear Authors, dear Reviewer QpC7,
>
> Thank you for deeply engaging into a discussion regarding the evaluation protocol for this paper. The precise way segmentation masks are obtained from the model(s) seems like important point that should likely not go unnoticed for future readers of the paper, and discussing and understanding the pros/cons of this protocol will likely provide significant value to the community.
>
> While following along, I was wondering whether the issue could be settled by including experiments on 1) a dataset that contains (uniformly) colored backgrounds and 2) greyscale or even binary multi-object datasets, but with otherwise similar visual complexity as the ones presented in the paper? A particularly well-suited dataset for 1) would be the Objects Room dataset which is available in the same library and format as the other datasets used in this paper, while for 2) the authors could consider adding experiments for greyscale CLEVR and binarized Multi dSprites as in the original Slot Attention paper [Locatello et al., 2020].
>
> I would be curious to hear both from Reviewer QpC7 and the authors what you think about the value of adding such experiments. I do not expect that the outcome of these experiment will affect the accept/reject decision of the paper, but I think it would be very valuable to have more clarity on this issue for future readers of the paper, independent of the outcome of the peer review process at NeurIPS.
>
> Thank you.
>
> --Your AC

---

> > ### Comment · Reviewer_QpC7 · 2023-08-18
> >
> > Dear Authors and AC, thank you for your replies.
> >
> > As far as I understand the presented evaluation protocoll, it may lead to issues whenever multi-channel inputs, such as RGB images, are used. Whenever these images contain pixels for which at least one channel has a value < 0.1, the evaluation procedure intertwines the separation of objects in the phases of the learned features with the objects' underlying color values.
> >
> > The authors have previously stated that this problem occurs very rarely in the multi-object datasets considered in the paper, stating that 0.44%, 0.88%, and 9.25% of pixels in CLEVR, dSprites, and Tetrominoes are affected. Since these numbers seemed suspiciously low to me, I ran my own calculations which paint a very different picture. Using the code below, I find that 0.9%, 12.6% and 100% of pixels in the training sets of CLEVR, dSprites, and Tetrominoes are affected. While these values might differ somewhat depending on the normalization used, this may render the results on the dSprites and Tetrominoes datasets uninterpretable. Even though some qualitative results show that the method can separate objects of the same color on the Tetrominoes dataset, it remains unclear how much of the remaining separation observed (between objects of varying colors and between objects and background) is actually due to the underlying model and how much is due to the underlying color values of the objects. As a result, I believe that, as it stands, only the results on CLEVR can be interpreted meaningfully.
> >
> > Regarding the AC's suggestion, I believe that the addition of another RGB dataset may only be helpful if that dataset is carefully designed to not contain pixels for which at least one channel has a value < 0.1. Alternatively, if the goal is to demonstrate the improved separability between objects and that the proposed method can represent more objects at the same time, a grayscale dataset could indeed be a good solution to circumvent the issues of the evaluation procedure.
> >
> >
> >
> > ---
> >
> > Code:
> >
> > 	# Assume images are in range [0, 1] and of shape [batch, channel, height, width].
> > 	low_magnitude_on_r = torch.stack(torch.where(images[:, 0] < 0.1))
> > 	low_magnitude_on_g = torch.stack(torch.where(images[:, 1] < 0.1))
> > 	low_magnitude_on_b = torch.stack(torch.where(images[:, 2] < 0.1))
> >
> > 	low_magnitude_on_rgb = torch.concat((low_magnitude_on_r, low_magnitude_on_g, low_magnitude_on_b), dim=-1)
> >
> > 	# Count each pixel only once, even when several channels are < 0.1.
> > 	pixels_with_low_magnitude = torch.unique(low_magnitude_on_rgb, dim=1).shape[1]
> > 	total_pixels = (images.shape[0] * images.shape[2] * images.shape[3])
> > 	low_magnitude_ratio = pixels_with_low_magnitude / total_pixels

---

> > > ### Author Response · Authors · 2023-08-18
> > > **Response to Reviewer and AC comments**
> > >
> > > Dear AC and reviewer, thank you for your engagement and for your constructive suggestions.
> > >
> > > Firstly, we would like to reply to the reviewers' suspicions about our calculations. In the initial review, the reviewer wrote:
> > > > (Reviewer QpC7 wrote): features with small magnitudes are masked out, … this rescaling may lead to a trivial separation of objects in RGB images. For an image containing a red and a blue object, for example, the reconstruction will assign small magnitudes to the color channels that are inactive for the respective objects.
> > >
> > > This states that the evaluation becomes an issue for “pure RGB colors” (as we called them in our first response), the ones that have one (or more) channel value *larger* than the threshold 0.1 and at least one value *smaller* than 0.1. The calculation that the reviewer presented here includes also the pixels for which *all* channels have a value below 0.1. These pixels are the black pixels in the image belonging mostly (or exclusively as in Tetrominoes) to the background. Therefore we think this calculation misses the point that we were all discussing about in the previous replies. As reviewer wrote, we were discussing about pixels that have “at least one channel has a value < 0.1” but also *not all channels have value <0.1*. In code, this correction boils down to the following change:
> > >
> > > ```
> > > pixels_with_low_magnitude = torch.unique(low_magnitude_on_rgb, dim=1).shape[1]
> > > pixels_with_all_channels_with_low_magnitude = low_magnitude_on_r * low_magnitude_on_g * low_magnitude_on_b
> > > total_pixels = (images.shape[0] * images.shape[2] * images.shape[3])
> > > low_magnitude_ratio = (pixels_with_low_magnitude - pixels_with_all_channels_with_low_magnitude) / total_pixels
> > > ```
> > >
> > > Having said all this, one last point on the percentages (although please let’s not switch the discussion back to this point now). Even by the reviewer’s calculation (which we don’t agree with) the percentages on dSprites and especially CLEVR dataset cannot account for the quantitative differences that are observed between the methods. Also for Tetrominoes, as the reviewer acknowledged, qualitative results show that the method is able to correctly group objects even in images of 2 or 3 objects of *exactly* the same color.
> > >
> > > Regarding the following suggestions from the AC:
> > > > (AC wrote): While following along, I was wondering whether the issue could be settled by including experiments on
> > > > (AC wrote): 1) a dataset that contains (uniformly) colored backgrounds (..Objects Room dataset..)
> > >
> > > Thank you very much for this suggestion. In general, dSprites and CLEVR used in our experiments already have a non-zero background but it is mostly grayscale, so the RGB-colored BG of Objects Room would expand on this. However, given the reviewers' worries that this dataset too might contain pixel values below 0.1 we are not sure what this experiment would add to the existing ones.
> > >
> > > > (AC wrote): 2) greyscale or even binary multi-object datasets (..greyscale CLEVR and binarized Multi dSprites..), but with otherwise similar visual complexity as the ones presented in the paper?
> > >
> > > > (Reviewer QpC7 wrote): if the goal is to demonstrate the improved separability between objects and that the proposed method can represent more objects at the same time, a grayscale dataset could indeed be a good solution to circumvent the issues of the evaluation procedure.
> > >
> > > In our architecture we contrast feature vectors both at the encoder and decoder outputs. However, in the grayscale images we could apply the contrastive objective only to the encoder output features which now would rely on edges, shape, texture and several other abstract visual features to guide the phase separation process. Though, we would be happy to run these experiments if it would be sufficient to convince the reviewer.

---

> > > > ### Comment · Area_Chair_XvoA · 2023-08-18
> > > >
> > > > Dear Authors,
> > > >
> > > > Thank you for the clarification. I agree that the Objects Room experiment I suggested would indeed not add much value as it would test for similar aspects as the datasets you already use, so feel free to ignore this suggestion. After reading both your responses, I think a Greyscale CLEVR experiment could still add significant value for readers of the paper, even though I don't think it would have an effect on the outcome of the review process for this submission (i.e. no need to rush this experiment).
> > > >
> > > > Thanks again,
> > > > --Your AC

---

> > > > > ### Comment · Reviewer_QpC7 · 2023-08-19
> > > > >
> > > > > Dear authors and AC, thank you for your replies.
> > > > >
> > > > > Since the paper evaluates the performance across all pixels when evaluating the separation between objects and background with the ARI-FULL score, I believe that black pixels should be equally considered when determining which pixels are affected by the faulty evaluation procedure.
> > > > >
> > > > > Overall, our discussion has led me to believe that the evaluation is even more problematic than I initially thought - I will therefore change my rating to a strong reject. Nonetheless, I would like to highlight that I believe the paper makes an important contribution to a relatively new and interesting research direction. The proposed contrastive loss makes elegant use of the magnitudes and phases present in the complex-valued features, and seems to improve object separability. Given a fixed evaluation procedure, I believe this would be a strong paper.

---

> > > > > > ### Author Response · Authors · 2023-08-19
> > > > > > **Results with zero threshold and without any threshold**
> > > > > >
> > > > > > A big thank you to the AC and reviewer QpC7 for your replies and engagement.
> > > > > >
> > > > > > To additionally support our argument that the decision to use a threshold of 0.1 does not affect our findings,  contributions and conclusions, we ran evaluation with two settings: one where the threshold is set to 0 (affects only pixels with *absolute zero* ([0,0,0]) output magnitude) and one setting **without any threshold**.
> > > > > >
> > > > > > From the results in the tables below we can conclude the following:
> > > > > >
> > > > > > 1. Using a threshold of 0 almost doesn’t change the ARI scores at all. In some cases it even improves the scores slightly: ARI-FG and ARI-FULL for CLEVR, ARI-FULL for dSprites and ARI-FG for CtCAE for Tetrominoes.
> > > > > >
> > > > > > 2. Using **no threshold at all** also marginally improves the ARI-FG and ARI-FULL scores on CLEVR and leads to almost identical (inside std-dev) results on dSprites.
> > > > > >
> > > > > > 3. Using no threshold at all leads to worse scores on Tetrominoes (though CtCAE still outperforms CAE++ with an even wider margin). However, this is not due to the drawback of the evaluation. This is due to an inherent “failure mode” of estimating the phase of a zero-magnitude complex number. The evaluation clusters the pixels based on their phase values. However, if a complex number has a magnitude of exactly zero, the phase estimation is ill-defined and will inherently be random. In practice, even for complex numbers that have a non-zero, but otherwise extremely small magnitude (e.g. on the order of 1e-5 or so) phase estimation will lead to inaccurate results. This “issue” with the noisy estimation of the phases of complex numbers with low magnitudes was already noted in Löwe et al. [1]: *”phase values of complex numbers with small magnitudes become increasingly random”*. In [1] the authors largely use (except for 1 experiment) datasets with absolute zero background. Not filtering those pixels in the evaluation would have led to completely different ARI scores, but more importantly *would have been wrong* because estimating the phase of a zero-magnitude complex number is ill-defined. Therefore, this is not a “failure” mode of our method or the evaluation, it is simply an “inherent difficulty” of working with complex numbers.
> > > > > >
> > > > > > Lastly, thank you again both for the discussion and insights that led us to analyze this aspect of our work too, which we believe added further value and clarity to it.
> > > > > >
> > > > > > A side note: even for (synthetic) datasets that have the extreme corner case as Tetrominoes, the above discussed “issue” could be avoided by, for example, appropriate image normalization, one that avoids dealing with pixels with zero values.
> > > > > >
> > > > > > [1] Löwe et. al, "Complex-Valued Autoencoders for Object Discovery'', TMLR 2022.
> > > > > >
> > > > > >
> > > > > > ---------------------------------------------------------------------------
> > > > > >
> > > > > >
> > > > > >
> > > > > >
> > > > > > | **Tetrominoes**   |                   |                   |
> > > > > > |-------------------|-------------------|-------------------|
> > > > > > | **Threshold=0.1** | **ARI-FG**        | **ARI-FULL**      |
> > > > > > | CAE++             | $ 0.78 \pm 0.07 $ | $ 0.84 \pm 0.01 $ |
> > > > > > | CtCAE             | $ 0.84 \pm 0.09 $ | $ 0.85 \pm 0.01 $ |
> > > > > > | **Threshold=0**   | **ARI-FG**        | **ARI-FULL**      |
> > > > > > | CAE++             | $ 0.77 \pm 0.07 $ | $ 0.79 \pm 0.02 $ |
> > > > > > | CtCAE             | $ 0.86 \pm 0.05 $ | $ 0.82 \pm 0.01 $ |
> > > > > > | **No threshold**  | **ARI-FG**        | **ARI-FULL**      |
> > > > > > | CAE++             | $ 0.54 \pm 0.09 $ | $ 0.21 \pm 0.05 $ |
> > > > > > | CtCAE             | $ 0.67 \pm 0.11 $ | $ 0.26 \pm 0.09 $ |
> > > > > >
> > > > > > -----------------------------------------------------------------------------
> > > > > >
> > > > > >
> > > > > > | **dSprites**      |                   |                   |
> > > > > > |-------------------|-------------------|-------------------|
> > > > > > | **Threshold=0.1** | **ARI-FG**        | **ARI-FULL**      |
> > > > > > | CAE++             | $ 0.38 \pm 0.05 $ | $ 0.49 \pm 0.15 $ |
> > > > > > | CtCAE             | $ 0.48 \pm 0.03 $ | $ 0.68 \pm 0.13 $ |
> > > > > > | **Threshold=0**   | **ARI-FG**        | **ARI-FULL**      |
> > > > > > | CAE++             | $ 0.38 \pm 0.05 $ | $ 0.49 \pm 0.12 $ |
> > > > > > | CtCAE             | $ 0.46 \pm 0.07 $ | $ 0.69 \pm 0.10 $ |
> > > > > > | **No threshold**  | **ARI-FG**        | **ARI-FULL**      |
> > > > > > | CAE++             | $ 0.37 \pm 0.05 $ | $ 0.49 \pm 0.12 $ |
> > > > > > | CtCAE             | $ 0.47 \pm 0.06 $ | $ 0.69 \pm 0.10 $ |
> > > > > >
> > > > > >
> > > > > >
> > > > > > -------------------------------------------------------------------------------
> > > > > >
> > > > > >
> > > > > >
> > > > > > | **CLEVR**         |                   |                   |
> > > > > > |-------------------|-------------------|-------------------|
> > > > > > | **Threshold=0.1** | **ARI-FG**        | **ARI-FULL**      |
> > > > > > | CAE++             | $ 0.22 \pm 0.10 $ | $ 0.30 \pm 0.18 $ |
> > > > > > | CtCAE             | $ 0.50 \pm 0.05 $ | $ 0.69 \pm 0.25 $ |
> > > > > > | **Threshold=0**   | **ARI-FG**        | **ARI-FULL**      |
> > > > > > | CAE++             | $ 0.33 \pm 0.04 $ | $ 0.32 \pm 0.25 $ |
> > > > > > | CtCAE             | $ 0.52 \pm 0.05 $ | $ 0.69 \pm 0.20 $ |
> > > > > > | **No threshold**  | **ARI-FG**        | **ARI-FULL**      |
> > > > > > | CAE++             | $ 0.34 \pm 0.04 $ | $ 0.32 \pm 0.25 $ |
> > > > > > | CtCAE             | $ 0.52 \pm 0.06 $ | $ 0.72 \pm 0.21 $ |

---

> > > > > > > ### Author Response · Authors · 2023-08-20
> > > > > > > **Apologies for the double posting of the above response**
> > > > > > >
> > > > > > > We posted the same response twice by mistake. We've deleted the second copy.

---

> > > > > > > ### Comment · Area_Chair_XvoA · 2023-08-21
> > > > > > >
> > > > > > > Dear reviewer QpC7,
> > > > > > >
> > > > > > > Could you please check whether the final response by the authors (which includes new experimental evidence for the case of not using any threshold during evaluation) addresses your concerns?
> > > > > > >
> > > > > > > Thank you,
> > > > > > > --Your AC

---

> > > > > > > > ### Comment · Reviewer_QpC7 · 2023-08-21
> > > > > > > >
> > > > > > > > Dear authors,
> > > > > > > >
> > > > > > > > Thank you for providing an improved evaluation procedure. I believe this largely addresses my concerns. I would like to ask the authors to adjust the evaluation in the rest of their paper accordingly, and to include an appropriate discussion of the potential drawbacks of using different thresholds. On top of this, it would also be interesting to see experiments on Tetrominoes with an improved normalization scheme that eliminates all-zero pixels.
> > > > > > > >
> > > > > > > > I still believe that an evaluation of the intermediate representations would be helpful, even if limited. Thus, I will increase my score to a weak accept.

---

> > > > > > > > > ### Author Response · Authors · 2023-08-21
> > > > > > > > >
> > > > > > > > > Dear reviewer QpC7 and AC,
> > > > > > > > >
> > > > > > > > > Thank you very much for the whole discussion which led us to analyze these aspects of our work, improve everyone’s understanding of evaluation of synchrony-based models and the rigor of our paper.
> > > > > > > > > We agree to adjust the evaluation in the rest of our paper as you suggested and to include a discussion of the potential drawbacks of using different thresholds. We will also run further experiments on Tetrominoes with an improved normalization scheme that eliminates all-zero pixels and attempt to evaluate the intermediate representations of our networks.
> > > > > > > > >
> > > > > > > > > Thanks to everyone for their time and engaging with us in these very productive discussions!
> > > > > > > > >
> > > > > > > > > All the best,
> > > > > > > > >
> > > > > > > > > Authors

---

### Author Rebuttal · Authors · 2023-08-09

### General Response to all Reviewers

We have provided individual responses to each reviewer's questions/comments. We have not made any changes to the PDF of the paper to ensure the ease of referring to the line numbers, Figure/Table numbers as in the original version. We’ve provided additional results (Tables) as part of the individual rebuttal responses. We’re happy to continue engaging with all reviewers to resolve any remaining concerns.

---

### Decision · Program_Chairs · 2023-09-21

**Decision:**

Accept (poster)

**Comment:**

This paper presents a collection of architectural improvements over Complex-Valued Autoencoders (CAEs) and introduces a novel contrastive objective to improve scene decomposition performance on a range of synthetic multi-object datasets. The architectural improvements and contrastive loss, while simple, provide clear benefits and allow this class of model to scale to more complex datasets (although still simplistic compared to real-world data).

All reviewers agree that this paper is very well-written and provides a clear advancement to an interesting architecture class that holds promise for learning structured scene representations. Even though this architecture class is still far from the performance of slot-based methods, the advancements are pointing in the right direction, and the paper should be of interest to the NeurIPS community.

There have been concerns around the evaluation protocol which the paper adopted, related to a thresholding heuristic that the authors chose to include (which was only described in the appendix) that could render the experimental comparison difficult to interpret. After an engaged discussion, the authors have sufficiently convinced the reviewers and myself that this heuristic does not have a large effect on the results. Nonetheless, it would be valuable to clearly highlight this aspect in the revised version of the paper.

Overall, this paper meets the bar for acceptance in my view and I am looking forward to seeing how future work will scale this class of models further to (hopefully) someday model real-world data to a convincing degree.